# DIVIDE-VERIFY-REFINE: ALIGNING LLM RESPONSES WITH COMPLEX INSTRUCTIONS

## ABSTRACT

Recent studies show that LLMs, particularly open-source models, struggle to follow complex instructions with multiple constraints, hindering their adoption in mission-critical applications. Despite the importance, methods to improve LLMs' adherence to such constraints remain largely unexplored, and current research focuses primarily on evaluating this ability rather than developing solutions. While a few studies enhance constraint adherence through model tuning, this approach is computationally expensive and heavily reliant on training data quality. An alternative is to leverage LLMs' self-correction capabilities, allowing them to adjust responses to better meet specified constraints. However, this self-correction ability of LLMs is limited by the feedback quality, as LLMs cannot autonomously generate reliable feedback or detect errors. Moreover, the self-refinement process heavily depends on few-shot examples that illustrate how to modify responses to meet constraints. As constraints in complex instructions are diverse and vary widely (e.g., text length, number of bullet points, or inclusion of specific keywords), manually crafting few-shot examples for each constraint type can be labor-intensive and sub-optimal. To deal with these two challenges, we propose the **Divide-Verify-Refine (DVR)** framework with three steps: (1) **Divide** complex instructions into single constraints and prepare appropriate tools; (2) **Verify**: To address the feedback quality problem, these tools will rigorously verify responses and provide reliable feedback (e.g., Python scripts for format checking or pre-trained classifiers for content analysis); (3) **Refine**: To address the constraint diversity challenge, we design a refinement repository that collects successful refinement processes and uses them as few-shot demonstrations for future cases, allowing LLMs to learn from the past experience during inference. Additionally, recognizing that existing datasets lack complexity and have internal conflict, we develop a new dataset of complex instructions, each containing 1-6 constraints. Experiments show that the framework significantly improves performance, doubling LLama3.1-8B's constraint adherence and tripling Mistral-7B's performance on instructions with 6 constraints. The code and dataset are available at https://anonymous.4open.science/r/CODE_ICLR2025-52CE/README.md.

## 1 INTRODUCTION

Large language models (LLMs), like ChatGPT, have shown significant improvements across a variety of language tasks (Ouyang et al., 2022; Touvron et al., 2023). The success of LLMs relies on their instruction-following ability to comprehend and execute complex instructions. Misinterpretations or failures to follow instructions can result in unintended outputs, which may have severe consequences (Mu et al., 2023; Zhou et al., 2023). This issue becomes particularly critical when LLMs are deployed as in high-stakes environments, such as legal documentation or technical writing. For example, when drafting legal contracts, LLMs must strictly adhere to constraints related to format, specific terminology, and precise language usage to avoid misinterpretations or legal liabilities. Similarly, in technical writing, adhering to strict format guidelines, word limits, and the inclusion of essential technical terms is critical to ensure clarity, consistency, and compliance with industry standards.

Recently, several studies show that LLMs, particularly open-source ones, struggle to follow complex instructions that contain multiple constraints, such as response length or formatting (He et al., 2024a;

Figure 1: (a) The LLMs hallucinate and cannot give reliable feedback. (b) The tools can check the response rigorously and provide reliable and directional feedback. The refinement repository provides past refinement examples and stores the current refinement process.

Jiang et al., 2024b; Chen et al., 2024b). Despite the recognition of this issue, research on enhancing LLMs' ability to follow constraints is still limited. Current efforts mainly focus on evaluating LLMs' constraint-following ability rather than on improving this ability (Jiang et al., 2024b; Chen et al., 2024b; Zhou et al., 2023). Only very few studies improve the LLMs' constraint-following ability through fine-tuning (He et al., 2024a; Sun et al., 2024; Li et al., 2024). Among them, one approach (He et al., 2024a) specifically enhances LLMs' ability to follow multiple constraints. It employs a teacher model to iteratively refine the outputs of a student model. This step-by-step correction process, along with the final accurate responses, is used to train the student model. Although fine-tuning is an effective approach, it usually requires a large amount of computation resources and heavily depends on the data quality. In contrast to training-based methods, the concept of "self-correction" offers an alternative approach, where LLMs autonomously correct their responses (Madaan et al., 2024; Shinn et al., 2024). Self-correction has been applied on various tasks such as question answering (Dhuliawala et al., 2023; Shinn et al., 2024) or mathematics (Madaan et al., 2024), where an LLM will evaluate its responses, give feedback, and further refine responses. For constraint-following, this self-correction process can be divided into two phases: verification and self-refinement (see Fig. 1(a)). During the verification phase, LLMs assess whether their responses align with the specified constraints. If the responses do not align with the constraints, the LLMs will give feedback that pinpoints errors and suggests adjustments. Following this, the self-refinement phase takes place where LLMs use the feedback to refine and improve their responses accordingly.

However, there are several challenges for an LLM to correct its response for multi-constraints. The first one is *feedback reliability*. Recent studies indicate that LLMs often exhibit only modest performance gains from self-correction, and the improvements can be unstable, occasionally even degrading performance in areas such as question answering (Huang et al., 2024) and code generation (Olausson et al., 2024). Several studies claim that the significant bottleneck in self-correction is the generation of reliable feedback (Tyen et al., 2024; Gou et al., 2024; Jiang et al., 2024a). LLMs, including advanced models like GPT-4 and Claude 3, tend to have low recall in detecting LLMs errors, underperforming significantly compared to humans (Kamoi et al., 2024). On the other hand, research reveals that the self-correction performance on reasoning tasks is boosted if the error location is given, indicating LLMs have the self-correction ability given reliable feedback (Tyen et al., 2024). From a constraint-following perspective, LLMs are also not good at checking simple and easy-to-verify constraints. As shown in Fig. 1(a), given a response, the LLMs struggle to accurately count the bullet points, sentences, or words. The second challenge is *constraint diversity* which lies in the self-refinement process. Given the response and the feedback, the LLMs should refine the response according to the feedback. However, to perform this task effectively, a set of representative few-shot examples is needed to demonstrate how to appropriately modify the response (Brown et al., 2020). These constraints can vary widely, from adhering to a length limit to including specific keywords. Each type of constraint needs distinct modifications. For example, meeting a length limit might require removing content, whereas incorporating specific keywords requires adding text. Manually crafting representative few-shot examples for each constraint type is labor-intensive.

To address these challenges, we propose a novel framework named Divide-Verify-Refine (DVR) as illustrated in Fig. 1(b). To enhance feedback reliability, we observe that constraints that LLMs struggle to verify, such as exact word counts, sentences counts, or bullet points, can be readily assessed using external tools. These tools include coding methods for quantitative measures, such as counting the number of words, sentences, paragraphs, or bullet points, as well as pre-trained

classifiers for content analysis, such as topic and sentiment analysis, which are easily accessible and widely available (Antypas et al., 2022; Loureiro et al., 2022). Instead of relying on LLMs to verify their responses, we instruct LLMs to interact with external tools to handle the verification process. We first instruct LLMs to divide the complex constraint into single constraints and then select one tool for each single constraint. By integrating these tools, we overcome the limitations of LLMs in verification and feedback, enabling more rigorous checking and providing reliable and detailed feedback for subsequent refinement. To address the constraint diversity problem, we propose the *refinement repository*. This repository is a memory module that collects and stores successfully refined examples for future use. Since the verification reliability is guaranteed by external tools, the successful refinement processes can be recorded and saved in the refinement repository. This allows LLMs to learn from past experiences during inference time.

Our **main contributions** are: (i) It is the first work to improve the LLMs' constraint-following ability without training. Our framework enhances feedback reliability by integrating easily accessible and widely available tools for verifying responses against specified constraints. (ii) To address the constraint diversity challenge, we propose a novel refinement repository to store successfully refined examples, which enables LLMs to learn from past experience and avoids crafting demonstrations manually. (iii) Most benchmarks only contain 1-2 constraints (Chen et al., 2024b). Current complex instruction datasets He et al. (2024a) combine seed instructions with external constraints but often overlook existing ones, causing conflicts and incomplete evaluations. To overcome the limitations and ensure a comprehensive evaluation, we construct a new complex instruction dataset with instructions containing 1-6 constraints.

## 2 RELATED WORK

**Instruction-Following of LLMs.** Since the ability to follow instructions is crucial for the practical use of LLMs, many recent studies evaluate this capability from various perspectives (Dubois et al., 2024; Zhou et al., 2023; Jiang et al., 2024b; Chen et al., 2024b; Zhou et al., 2023; He et al., 2024b). They evaluate LLMs' instruction-following ability by testing on length (Dubois et al., 2024), format (Zhou et al., 2023), semantic and topic constraints (Chen et al., 2024b). Most works only test LLMs on simple instructions with only 1-2 constraints. Recently, some works generate instructions with multiple constraints (He et al., 2024a; Jiang et al., 2024b). They find that LLMs struggle to follow complex instructions as the number of constraints increases. Moreover, there is a big performance gap between the open-source models and the closed-source models on instruction-following. Upon finding these problems, some seminal works aim to improve instruction-following the ability of LLMs (Chen & Wan, 2023; Sun et al., 2024; Wang et al., 2024; He et al., 2024a). They use various prompting strategies to generate instructions and responses with advanced LLM models (e.g., GPT4) and then use the generated data to fine-tune open-source LLMs. Though most methods only consider instructions with few constraints, one of them (He et al., 2024a) focuses on improving the LLMs' ability to follow multiple constraints. They generate complex instruction datasets by merging instructions with external constraints. Then, they adopt GPT4 (teacher model) to modify the response of the open source model (student model) iteratively. The student model is finetuned by both the intermediate modification process and the final modified response. Although fine-tuning is an effective approach, it usually requires a large number of computation resources and heavily de- pends on the data quality. In addition, the fine-tuned model will still suffer from new constraints not seen before. Different from previous methods, our framework uses in-context learning with tool interaction to effectively identify and rectify unsatisfactory responses using the LLM itself, which is more practical and accessible.

**Self-Correction of LLMs.** Self-correction is a framework where LLMs refine their own responses during inference by reflecting on their initial responses (Shinn et al., 2024; Madaan et al., 2024). This process can be divided into two phases. Initially, LLMs are prompted to analyze and provide feedback on their own responses. Subsequently, based on the feedback LLMs refine the responses to correct their mistakes. However, recent studies report negative results indicating that LLMs cannot self-correct their own mistakes (Hong et al., 2024; Tyen et al., 2024; Kamoi et al., 2024; Gou et al., 2024). For example, Kamoi et al. (2024) reveal that top LLMs like GPT-4 and Claude 3 have low recall in detecting LLM errors, with LLMs significantly underperforming compared to humans. Additionally, feedbacks provided by LLM self-correction tend to hallucinate and lack reliability. This unreliability suggests that even when errors are detected, the guidance offered for corrections

Figure 2: The DVR framework: (a) Divide: The LLMs decompose constraints and instantiate tools for each constraint, (b) Verify: Tools will give feedback on the response, (c) Refine: The refinement repository provides past refinement process as few-shot examples. The current refinement process will be stored in the repository.

may be incorrect or misleading. Hong et al. (2024) find that LLMs struggle to accurately identify logical fallacies, casting doubt on their inherent ability to detect errors and conduct self-verification reasoning effectively. However, the self-correction performance on reasoning tasks is boosted if the error location is given (Tyen et al., 2024). A study also shows that LLMs do have the ability to correct their responses with the help of tools (Gou et al., 2024). All these observations indicate that LLMs themselves are not reliable in analyzing their responses and a more reliable feedback mechanism is needed to pinpoint the mistakes. The most closely related work is CRITIC (Gou et al., 2024), which uses tools for verification, such as an external API for testing the toxic score of the response. Our method differs from CRITIC by incorporating a refinement repository module that accumulates successful past refinements, thereby enhancing the effectiveness of self-correction over time. Moreover, we use multiple tools to provide a detailed analysis of various aspects of the response and give modification suggestions, instead of giving a single toxic score.

**LLMs Using Tools.** Tools have been extensively employed to enhance the capabilities of LLMs across various domains. For instance, retrievers are used to augment response generation of LLMs by fetching relevant information (Khandelwal et al., 2019), while search engines enhance the model's access to real-time data Nakano et al. (2021). Similarly, calculators are adopted to support math reasoning of LLMs (Cobbe et al., 2021), interpreters are used to facilitate accurate code generation (Chen et al., 2022; Gao et al., 2023), and mathematical provers help in verifying theoretical proofs (Jiang et al., 2023). We observe that following complex instructions can be hard, but checking them with tools is much easier. Because of this, we integrate external tools to our LLMs to help with this process. This integration can provide reliable verification and detailed feedback and also enable the LLMs to save past refinement examples for future use.

## 3 THE PROPOSED FRAMEWORK: DVR

As shown in Fig. 2, we propose the *Divide-Verify-Refine* (DVR) framework which consists of three modules: (a) *Divide* instructions and prepare tools accordingly, (b) *Verify* responses and provide feedback, and (c) *Refine* and store responses in a repository. First, The tool preparation module aims to identify constraints, select appropriate tools, and fill out parameters. In this module, LLMs first decompose the complex instructions into single constraints. For each single constraint, the LLMs will prepare appropriate tools for verification. Second, in the verification and feedback module, the prepared tools will verify the response and give detailed feedback if the response does not adhere to the constraint. Third, in the self-refinement module, given the feedback and past refinement experience for the same constraint as few-shot examples, the response is refined to adhere to the target constraint. The successfully refined response will be stored in the refinement repository so that it can be retrieved as refinement examples in the future. Next, we introduce each model in detail.

### 3.1 DIVIDE: TOOL PREPARATION

To provide accurate feedback, we propose to adopt tools for verification. To enable LLMs to use external tools, we construct tools for different types of constraints. We build Python verifiers for

structural constraints (e.g., the number of words, the number of paragraphs, or the number of bullet points). To handle constraints where existing tools are unavailable, we can request advanced code-generation models such as DeepSeek-Coder (Zhu et al., 2024) or Code-Llama Roziere et al. (2023) to generate required tools. We also adopt existing classifiers as verifiers for content constraints (e.g., the topic classifier (Antypas et al., 2022) and the sentiment classifier (Loureiro et al., 2022)).

Given an input instruction $I$, the LLM $\mathcal{M}$ first decomposes it into a series of individual constraints. We use a decomposition prompt $p_{decomp}$ asking LLMs for decomposition. With input instruction and decomposition prompt, LLM then generates a set of decomposed constraints: $\mathcal{M}(p_{decomp}, I) \rightarrow \{c_i\}_{i=1,2,3...}$, where $c_i$ is the $i$-th single constraint. For each constraint $c_k$, the LLM determines the appropriate tool by matching $c_k$ to a tool $t_k$ from the predefined toolset: $\mathcal{M}(p_{select}, c_k) \rightarrow t_k$, where $t_k \in \{t_i\}_{i=1,2,3...}$ is the selected tool for the constraint $c_k$. The prompts for decomposition $p_{decomp}$ and tool selection $p_{select}$ are in Table 20 in Appendix A.13. After selecting the tools, the LLM sets the necessary parameters for each tool, such as specifying the required number of bullet points or the desired sentiment for the response. Finally, all tools relevant to instruction $I$ are compiled into the set $T_I = \{t_i\}_{i=1,2,3...}$, ready to be utilized in the subsequent verification and feedback phase.

## 3.2 VERIFY: VERIFICATION AND FEEDBACK

Given the instruction, the LLM will first generate the initial response $R_0 = \mathcal{M}(p_{generate}, I)$, where $p_{generate}$ is the prompt for generation (detailed in Appendix A.13). We denote the current response as $R$ and $R = R_0$ for the first round of refinement and will be updated to the refined response in subsequent rounds. The current response is verified by each tool in toolset $T_I$ as follows:

$$f_i = t_i(R), \forall t_i \in T_I \tag{1}$$

where $f_i$ is the feedback from tool $t_i$ for constraint $c_i$. If the response adheres to the constraint, the feedback is a boolean value "true". Otherwise, $f_i$ is a textual feedback that first identifies the error in the response and then suggests modification. For example, as shown in Fig. 2, the tool "Bullet_points(4)" counts the number of bullet points in the response and outputs "true" if there are 4 bullets; while the response only contains 2 bullets. It finds that the response does not satisfy the constraint and gives out the feedback "The response only contains 2 bullet points. 2 more bullet points should be added." This detailed feedback points out the errors in the response and gives directional information for LLMs to modify the response. We collect all feedback $F_I = \{f_i\}_{i=1,2,3...}$ which will be used to refine the response $R$.

## 3.3 REFINE: SELF-REFINE WITH FEEDBACK AND FEW-SHOT DEMONSTRATION

In the self-refinement phase, the LLM leverages the feedback collected to refine the response. To improve the performance, we propose to adopt representative demonstrations in the prompt to instruct LLMs on how to conduct refinement using feedbacks. As constraints vary widely, each type of constraint requires specific demonstrations for effective refinement. Manually creating few-shot examples for each constraint type is labor-intensive and impractical for real-world applications. To solve this issue, we propose to store the successful refinement process in the refinement repository. When LLMs need to refine a new response involving the same constraint type, the few-shot examples can be retrieved from the refinement repository as in-context examples.

Specifically, the refinement process targets one unsatisfied constraint at a time, cycling through a refine-verify-refine loop until all constraints are satisfied. For a given response $R$ and the feedback $f \in F_I, f \neq True$, the refinement response can be written as follows:

$$R' = \mathcal{M}(p_{refine}, s^t, I, R, f) \tag{2}$$

where $p_{refine}$ is the prompt for refinement (detailed in Appendix A.13), $s^t = \{(I_i, R_i, f_i, R'_i)^t\}_{i=1,2,3...}$ is the set of refinement examples selected from the refinement repository $Q$, which contains refinement examples having the same constraint type associated with $f$. There might be many refinement examples having the same constraint type with $f$ available in the refinement repository. Retrieval techniques like semantic similarity can be employed to select the most relevant examples. In this paper, we randomly select relevant examples for simplicity and leave more advanced techniques as future work. Some refinement examples are in Table 18 and 19 in Appendix A.12.

If the refined response adheres to the constraint, i.e., $t(R') = True$, the current successful refinement process will be stored in the repository as $Q = Q \cup \{(I, R, f, R')^t\}$.

**Discussion.** Our method, DVR introduces a novel approach to enhancing LLMs' ability to follow complex instructions with multiple constraints. The detailed algorithm of DVR is shown in Algorithm 1 in Appendix A.1. By integrating external tools for reliable and detailed feedback and a refinement repository for storing successful refinement examples, we provide a scalable and robust framework for improving instruction compliance without the need for extensive retraining. Moreover, the external tools and the refinement repository work jointly. Without reliable feedback, the refinement repository would risk accumulating incorrect or noisy examples, which could deteriorate the performance of LLMs over time. The detailed feedback gives "directional" information, which guides the LLMs to adjust their responses. Compared to directly following complex instructions, decomposing these instructions and selecting the appropriate tools are simpler tasks for LLMs. This inherent advantage allows our framework to be very effective, as it leverages these easier tasks to build a robust system that enhances the LLMs' adherence to constraints.

## 4 EMPIRICAL VALIDATION

In this section, we conduct experiments to answer the following research questions: (**RQ1**) Can our DVR improve the ability of LLMs to follow complex constraints? (**RQ2**) How does the performance of LLMs differ across various types of constraints, and which constraints pose the greatest challenges? (**RQ3**) How does each module of DVR (the tool-assisted verification and the few-shot self-refinement library) individually contribute to improving LLMs' ability to follow constraints?

### 4.1 EXPERIMENTAL SETUP

**Datasets.** We conduct experiments on two datasets: (i) We conduct experiments on **CoDI** (Controllable Generation under Diversified Instructions) (Chen et al., 2024b). It has 500 instructions with 2 constraints. Each instruction has a topic constraint and a sentiment constraint. (ii) **ComplexInstruct**: Since the complexity of CoDI is limited, we construct a new complex instruction datasets called ComplexInstruct. We use CoDI (topic instruction set) (Chen et al., 2024b) as seed instructions which ask users to generate text on certain topics. Some instructions ask users to generate "a paragraph of..." or "a sentence of..." which already contain length constraints. To avoid conflicts and hidden constraints, we remove these constraints by replacing the keywords "paragraph" and "sentence" with "text". Then, we synthesize complex instructions by adding constraints to these seed instructions (Zhou et al., 2023). To simulate instruction of different levels, we generate 6000 complex instructions with 1-6 constraints for each instruction as 6 levels (1000 instructions for each level). We have 21 types of constraints categorized into 8 general categories (such as length constraint, punctuation, and case change). Each type of constraint is diversified into 8 different expressions. The detailed information about the constraint types is in Appendix A.3 and Table 6.

**Baselines.** We compare our method with representative and state-of-the-art baselines, which can be categorized into three main types: (i) Self-reflection based methods, which iteratively improve response via feedback from LLMs reflection, such as Reflexion (Shinn et al., 2024); (ii) Prompting based methods, which use different prompting strategies to get the best response, including Branch-solve-Merge (BSM) (Saha et al., 2024) and Universal Self-Consistency (U-SC) (Chen et al., 2024a); and (iii) Tool based methods, which use external tools for feedback or selection, such as Rejection sampling (Saunders et al., 2022), React (Yao et al., 2023), and CRITIC (Gou et al., 2024). For the refinement repository of our framework, we consider two variants, i.e., warm-start and cold-start. For warm-start, we have an additional set of instructions (6000 samples for ComplexInstruct and 500 samples for CoDI). Note that these data samples are totally independent with test set. Our framework will first run on these samples to collect examples to fill the refinement repository. For cold-start, since the refinement repository is empty at beginning, we use 5 fixed few-shot examples if there are no examples that can be retrieved from the repository. The detailed information about each baseline is in Appendix A.2.

**Implementation.** We test on popular open-source models including Mistral-7B, Llama3-8B, Llama3.1-8B and Llama3.1-70B. The temperature of the model is 0.8. We set the number of few-shot demonstrations for initial response generation and self-refinement (without repository) as 5 for

Table 1: Instruction Satisfaction Rate (ISR) across levels 1 to 6 (Llama-3.1-8B-Instruct). The values in parentheses ($+xx$) indicate the improvement compared to the best performing baseline.

| Method | Level 1 | Level 2 | Level 3 | Level 4 | Level 5 | Level 6 |
|---|---|---|---|---|---|---|
| Vanilla | 90.5 | 76.6 | 62.5 | 50.1 | 35.6 | 25.3 |
| Reflxion | 91.6 | 78.1 | 63.7 | 49.8 | 35.8 | 25.7 |
| BSM | 90.1 | 75.3 | 62.0 | 47.5 | 35.5 | 24.1 |
| U-SC | 90.9 | 76.3 | 62.4 | 47.1 | 36.0 | 25.8 |
| Rejection Sampling | 92.1 | 86.7 | 71.1 | 60.4 | 49.8 | 36.3 |
| ReAct | 94.2 | 86.1 | 72.5 | 60.7 | 50.2 | 37.2 |
| CRITIC | 93.8 | 87.1 | 75.4 | 64.4 | 52.4 | 43.2 |
| DVR (coldstart) | 94.5 (+0.7) | 87.9 (+0.8) | 78.4 (+3.0) | 69.5 (+5.1) | **60.9** (+8.5) | 49.2 (+6.0) |
| DVR (warmstart) | **95.2** (+1.4) | **88.7** (+1.6) | **79.2** (+3.8) | **69.7** (+5.3) | 60.5 (+8.1) | **49.6** (+6.4) |

Table 2: Instruction Satisfaction Rate (ISR) across levels 1 to 6 (Mistral-7B-Instruct-v0.3)

| Method | Level 1 | Level 2 | Level 3 | Level 4 | Level 5 | Level 6 |
|---|---|---|---|---|---|---|
| Vanilla | 77.0 | 55.3 | 34.1 | 19.9 | 12.4 | 6.3 |
| Reflxion | 77.2 | 55.8 | 35.1 | 20.1 | 12.0 | 5.8 |
| BSM | 78.1 | 56.2 | 33.8 | 19.3 | 11.3 | 5.2 |
| U-SC | 76.8 | 56.0 | 34.3 | 20.4 | 12.9 | 5.8 |
| Rejection Sampling | 78.4 | 58.3 | 37.6 | 23.0 | 13.5 | 6.8 |
| ReAct | 86.0 | 67.8 | 46.0 | 32.5 | 18.2 | 10.7 |
| CRITIC | 88.9 | 72.5 | 55.6 | 43.5 | 28.1 | 18.1 |
| DVR (coldstart) | 94.9 (+6.0) | 80.2 (+7.7) | 64.1 (+8.5) | 49.3 (+5.8) | 35.8 (+7.7) | **23.6** (+5.5) |
| DVR (warmstart) | **95.0** (+6.1) | **81.3** (+8.8) | **66.6** (+11.0) | **51.4** (+7.9) | **36.4** (+8.3) | 23.4 (+5.3) |

our method and every baseline. We use the same set of few-shot demonstrations both for baselines and our method. We also set the maximum number of few-shot demonstrations for refinement (with repository) as 8. We set the number of trials as 5 for our method and every baseline.

**Evaluation Metrics.** We assess the constraint-following ability by calculating the Instruction Satisfaction Rate (ISR) (Jiang et al., 2024b). Specifically, each single instruction is satisfied when all constraints in that instruction are satisfied. It is calcualted as ISR = $\frac{1}{N} \sum_{i=1}^{N} \prod_{j=1}^{m_i} c_{ij}$, where $N$ is the total number of instructions in the dataset, $m_i$ is the number of constraints in the $i$-th instruction, $c_{ij} = 1$ if the $j$-th constraint in $i$-th instruction is satisfied; otherwise $c_{ij} = 0$.

## 4.2 RQ1: ASSESSING THE CONSTRAINT-FOLLOWING ABILITY

To answer RQ1, we evaluate our framework on two datasets. We evaluate structural constraints (e.g., text length, number of sections, and bullet points) on ComplexInstruct and content constraints (e.g., topic and sentiment constraints) on CoDI respectively.

For ComplexInstruct, there are six difficulty levels. Each level corresponds to the number of constraints in the instruction. For example, each instruction in Level 3 contains three constraints. There are 1000 instructions for each level. Results are shown in Table 1 and Table 2 (more results in Appendix A.4). **(i) Single constraints vs Multi-constraints:** For instructions in different difficulty levels, responses to instructions with more constraints tend to have a lower satisfaction rate. The satisfaction rate of instructions at Level 1 can approach close to 100%. However, at Level 6, the satisfaction rates for models like Llama3.1-8B and Mistral-7B drastically fall to 25% and 6.3%, respectively. This indicates that LLMs struggle to satisfy instructions with multiple constraints even though LLMs can satisfy them individually. **(ii) Self-Reflection is unreliable:** We can observe that Reflxion (Shinn et al., 2024) where LLMs reflect on and self-correct their responses, provides little improvement over Vanilla. This result indicates that LLMs themselves can not effectively identify their errors in constraint-following tasks. Similarly, Universal Self-consistency (Chen et al., 2024a), which allows LLMs to choose the most consistent answers from a set of candidates, has small improvements over Vanilla. These observations indicate that self-reflection and conventional prompting techniques may not sufficiently enhance LLMs' ability to follow constraints, highlighting the potential need for external tools to assist LLMs in this area. **(iii) Tools are helpful:** ReAct and CRITIC can be viewed as two variants in our framework. In contrast, CRITIC offers more granular feedback by specifically identifying which constraint within an instruction is not satisfied.

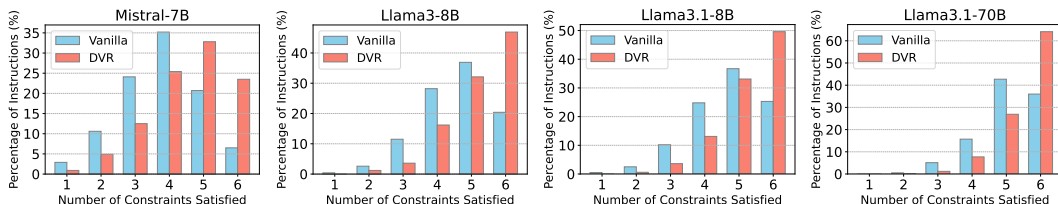

Figure 3: Distribution of satisfied constraints number per instruction (level 6).

Table 3: Instruction Satisfaction Rate (ISR) on CoDI dataset

| Method | Mistral 7B | Llama3 8B | Llama3.1 8B |
|---|---|---|---|
| Vanilla | 68.8 | 68.8 | 68.6 |
| Reflexion | 69.4 | 70.0 | 69.8 |
| Branch-Solve-Merge | 68.2 | 68.4 | 68.6 |
| Universal Self-consistency | 69.2 | 70.2 | 69.6 |
| Rejection Sampling | 79.8 | 80.8 | 81.4 |
| ReAct | 80.4 | 81.0 | 81.8 |
| CRITIC | 88.6 | 93.0 | 91.2 |
| DVR (coldstart) | 92.0 | 94.2 | **94.6** |
| DVR (warmstart) | **93.2** | **94.4** | **94.6** |

Compared with ReAct Yao et al. (2023), CRITIC Gou et al. (2024) has better performance, which means even identifying the unsatisfied constraint can also be helpful. This extra information helps the LLMs to locate the error when the number of constraints increases. Our method outperforms all baselines and the performance gain is larger for more complex instructions (level 4 to level 6).

By comparing the performance between different models, the larger model (Llama3.1-70B) performs better than the smaller models (Llama3.1-8B, Llama3-8B, and Mistral-7B). Additionally, with the same model size, Llama3.1-8B performs better than Llama3-8B in constraint-following.

The results in Figure 3 illustrate the distribution of constraints satisfied per instruction at the level 6 difficulty. There is a shift in the distribution towards the right when using our framework, indicating an enhancement in the ability to meet multiple constraints. Specifically, for the Mistral-7B model, the implementation of our framework shifts the center of the distribution from satisfying 4 constraints to satisfying 5 constraints.

We also observe that LLMs perform better on the CoDI dataset Chen et al. (2024b). There are two reasons. The first reason is that instructions are relatively simple, and only contain two constraints. Additionally, another study also shows that LLMs perform relatively better on sentiment and topic constraints Chen et al. (2024b) compared with format constraints. The LLMs inherently have better performance on semantic constraints over structural constraints. Our methods also outperform baselines and successfully improve the instruction satisfaction rate on CoDI.

### 4.3 RQ2: COMPARISON ACROSS DIFFERENT CONSTRAINT TYPES

Comparison across different constraint types is shown in Table 4 (warmstart). Coldstart results are provided in Table 9 in Appendix A.4. The 21 constraints in ComplexInstruct are categorized into 8 general categories as shown in the table. We have the following observations. (i) We can find that length constraints are the most challenging constraints for every language model. Length constraints have three levels: a minimum or maximum word count, a minimum or maximum sentence count, and an exact number of paragraphs. The reason might be that there is a lack of instructions containing length constraints during the instruction-tuning process. As a result, the language model struggles to understand the relationship between the output and the specified length in the instruction. Moreover, LLMs must plan from the beginning of the generation process to not only meet the length constraint but also ensure that the response remains complete and coherent. (ii) Language constraints, which require the use of languages such as Italian, German, or Japanese, are the second most challenging. This might be due to the limited multilingual capabilities of the LLMs. (iii) Punctuation constraint which requires LLMs not to use any commas in their responses, is especially challenging for Mistral-7B. However, our framework improves it significantly and triples the performance from 24% to

Table 4: Comparison Across Different Constraints Types

| Constraint Type | Mistral-7B | | Llama3-8B | | Llama3.1-8B | | Llama 3.1-70B | |
|---|---|---|---|---|---|---|---|---|
| | Vanilla | DVR | Vanilla | DVR | Vanilla | DVR | Vanilla | DVR |
| Detectable Content | 76.36 | 88.90 | 84.18 | 96.81 | 86.29 | 96.31 | 97.06 | 98.59 |
| Keywords | 76.04 | 84.23 | 83.84 | 88.32 | 84.94 | 88.77 | 87.88 | 92.03 |
| Punctuation | 24.34 | 72.93 | 91.03 | 95.64 | 97.01 | 98.04 | 98.38 | 98.39 |
| Case Change | 70.08 | 81.28 | 81.28 | 93.38 | 82.97 | 90.71 | 80.23 | 96.28 |
| Start End | 81.29 | 90.41 | 84.88 | 90.03 | 84.07 | 91.92 | 89.37 | 96.46 |
| Detectable Format | 69.59 | 80.70 | 81.57 | 89.23 | 84.69 | 92.31 | 90.52 | 95.30 |
| Language | 69.11 | 81.80 | 77.06 | 88.38 | 81.96 | 89.76 | 90.83 | 95.26 |
| Length Constraints | 50.42 | 73.23 | 65.29 | 80.85 | 68.55 | 83.57 | 80.50 | 90.20 |

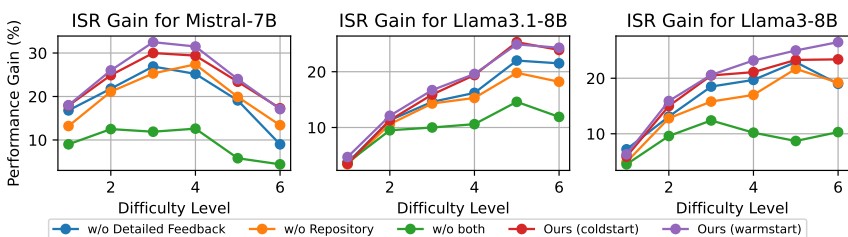

Figure 4: Ablation study on Mistral-7B, Llama3.1-8B and Llama3-8B.

73% satisfaction rate. The reason might be that Mistral 7B tends to ignore this constraint and needs external feedback to correct the answer.

## 4.4 RQ3: Contribution of Individual Modules

In addition to comparing our method with existing baselines, we conduct an ablation study to assess the effectiveness of individual modules. Specifically, we investigate three variants: (i) **w/o Detailed Feedback:** The detailed feedback from the tool is removed but the refinement repository is kept to provide relevant few-shot examples showing the responses before and after refinement. Here, the refinement repository is empty in the beginning (coldsart). (i) **w/o Repository:** The refinement repository is removed, and only 5 fixed examples are used for the self-refine process. (i) **w/o both:** The refinement repository and detailed feedback are all removed. Tools only give whether the whole instruction is satisfied. Figure 4 shows performance gaps between each method and the Vanilla. Both detailed feedback and the refinement repository are crucial. Without the repository, performance gains are limited, as fixed few-shot examples aren't optimal for each refinement target. Detailed feedback is also important for LLMs, because it locates the error and provides the direction for LLMs to modify their responses. Llama3-8B and Llama3.1-8B show higher gains on more difficult instructions. Mistral-7B's gains are modest, because of the limited capacity of Mistral-7B in following complex instructions, it can only follow 6.3% of level 6 instructions (shown Table 2). Despite this, the gains are notable from its low starting point. In conclusion, both detailed feedback and the refinement repository contribute to the performance of our framework.

## 4.5 Hyper-Parameter Sensitivity Analysis

We also conduct a hyper-parameter sensitivity analysis of our framework, testing different numbers of refinement few-shots and trials for successful refinement on Llama3.1-8B. As shown in Figure 5, performance improves with more trials but saturates at five, with minimal gains beyond that. Similarly, increasing the few-shot examples boosts performance in the beginning. The performance saturates after 8 shots. On the CoDI dataset, performance improves rapidly with initial increases in trials and examples, indicating that the first few numbers of trials and few-shot examples are most effective for refining the response.

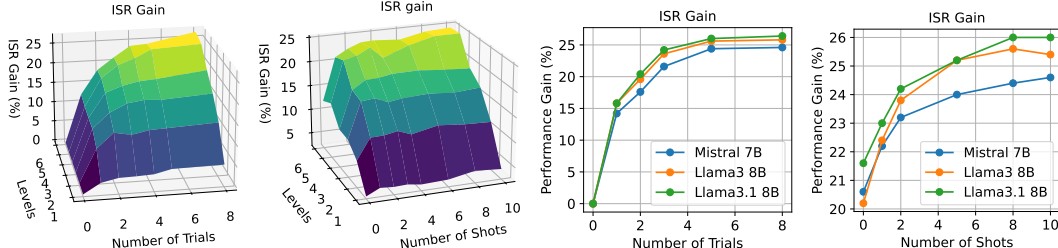

Figure 5: Parameter study on ComplexInstruct (first two charts) and CoDI dataset (last two charts).

Table 5: LLMs Performance on Tool Selection (%)

| Models | Hamming Loss | Accuracy | Precision | Recall | F1 Score |
|---|---|---|---|---|---|
| Mistral-7B | 4.13 | 52.85 | 92.98 | 81.39 | 86.80 |
| Llama3-8B | 2.64 | 67.60 | 94.39 | 89.48 | 91.87 |
| Llama3.1-8B | 2.90 | 61.77 | 94.69 | 87.50 | 90.95 |
| Llama3.1-70B | 0.86 | 86.40 | 98.41 | 96.38 | 97.38 |

## 4.6 TOOL SELECTION ACCURACY

Correctly decomposing and selecting tools are essential for feedback and refinement. We define tool selection as a multi-label prediction task for LLMs, evaluated using hamming score, accuracy, precision, recall, and F1-score. The total number of tools is 21. Results are shown in Table 5. Hamming loss, which measures the fraction of incorrect labels, is low across all models, indicating minimal mispredictions. Every model demonstrates a very high precision score, meaning that the tools they select are mostly correct, avoiding misleading feedback with incorrect tool selection. Accuracy, which measures the exact match between the selected tools and the ground truth, is the strictest metric. Despite this, all models achieve over 50% accuracy. Considering the limited performance of these models on constraint-following tasks, tool selection is a relatively easier task for LLMs. This performance gap makes it possible for our method to provide reliable feedback, collect past refinement examples and be effective in improving LLMs' constraint-following ability.

## 4.7 WOULD DVR AFFECT COMPREHENSIBILITY AND FLUENCY OF RESPONSES

In this subsection, we investigate if our framework would sacrifice comprehensibility and fluency in order to follow complex-constraints. We focus on evaluating key metrics such as readability, perplexity, and coherence. These metrics assess the comprehensibility and fluency of the responses. Results on ComplexInstruct (Table 11) and CoDI (Table 12) are in Appendix A.6. They both show that our framework has performance comparable to those of Vanilla, indicating that it does not degrade fluency and readability. The reason is that our method does not change any weights in LLMs, which maintains their ability in generating fluent and comprehensible text.

## 5 CONCLUSION

In conclusion, this paper presents the Divide-Verify-Refine (DVR) framework to enhance LLMs' ability to follow multi-constraint instructions. There are three steps in our framework: (1) Divide complex instructions into single constraints and assign appropriate tools for each constraint. (2) Verify: To deal with the feedback quality problem, these tools rigorously verify the response and generate reliable feedback. (3) Refine: To tackle the constraint diversity challenge, we design the refinement repository to store successful refinement processes, allowing LLMs to retrieve and learn from past examples. Our framework improves LLMs' adherence to complex multi-constraint instructions without the need for retraining, offering a scalable solution to enhance the practical usability of LLMs in real-world applications. Additionally, we construct a new dataset free from hidden or conflicting constraints, providing a more comprehensive and accurate evaluation of LLM performance on multi-constraint instructions.

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

# A   APPENDIX

## A.1   ALGORITHM

---

**Algorithm 1** Algorithm for DVR

---

**Input:** Language Model $\mathcal{M}$, Input Instructions $X$, Toolset $T$
**Output:** Response set $Y$
**Select:** Number of trials $n$

1: Initialize the refinement repository $Q = \{\}$
2: **for** $I \in X$ **do**
3:    Generate initial response: $R_0 = \mathcal{M}(p_{generate}, I)$.
4:    Initialize the toolset for instruction $I$: $T_I = \{\}$.
5:    Decompose constraints: $\mathcal{M}(p_{decomp}, I) \rightarrow \{c_i\}_{i=1,2,3...}$
6:    **for** $c \in \{c_i\}_{i=1,2,3...}$ **do**
7:       $\mathcal{M}(p_{select}, c) \rightarrow t$, where $t \in T$.
8:       $\mathcal{M}$ sets parameters for $t$.
9:       $T_I = T_I \cup t$.
10:   **end for**
11:   $R = R_0, a = n$.
12:   **while** $a > 0$ **do**
13:      $a = a - 1$
14:      Verify and get feedback from tools: $F_I = \{f_i\}_{i=1,2,3...}$, where $f_i = t(R)$.
15:      **if** $f = True, \forall f \in F_I$ **then**
16:         **return** $R$
17:      **end if**
18:      Retrieve few-shot examples: $s^t = \{(I_i, R_i, f_i, R_i')^t\}_{i=1,2,3...}$, where $s^t \subseteq Q$.
19:      Refine: $R' = \mathcal{M}(p_{refine}, s^t, I, R, f)$, where $f \in F_i$ and $f \neq True$.
20:      **if** $t(R') = True$ **then**
21:         Save the refinement process: $Q = Q \cup \{(I, R, f, R')^t\}$
22:         Update the current response: $R = R'$
23:         $a = n$
24:      **end if**
25:   **end while**
26:   $Y = Y \cup R$
27: **end for**

---

## A.2   BASELINE DETAILS

- Reflexion (Shinn et al., 2024): This method allows LLMs to self-reflect on their own responses and provide valuable feedback for future outputs. With the feedback, LLMs will refine their responses.

- Branch-solve-Merge (BSM) (Saha et al., 2024): BSM uses a "Divide and Conquer" approach to break complex instructions as individual branches. Then the LLMs will merge the responses from branches as the final answer. Similarly, in our experiment, we use LLMs to generate a response for each single constraint and then merge them together.

- Universal Self-Consistency (U-SC) (Chen et al., 2024a): This study extends the idea of Self-Consistency (Wang et al., 2023) to free-form generation. It first generates several candidate responses and then asks LLMs to select the most consistent one.

- Rejection Sampling (Saunders et al., 2022): Since we have tools for reliable verification, the most simple method is to select the best one from a set of responses. Here, we give the maximum number of trials as 5.

- ReAct (Yao et al., 2023): In ReAct, LLMs take actions based on the observation of the environment. Here, we adopt this method by letting the tools as the environment and giving LLMs boolean signals indicating whether the generated response adheres to all constraints in the instruction.

- CRITIC (Gou et al., 2024): CRITIC uses tools for verification such as using external API for evaluating the toxic score of the response or code executor for checking code generation. We adopt this method into our scenario, where tools will pinpoint which constraint of the instruction is not satisfied.

## A.3 COMPLEXINSTRUCT

We have 21 types of constraints which can be divided into 8 general categories (Zhou et al., 2023):

- Keywords:
  (1) Include keyword,
  (2) Include keyword at least/less than certain frequency,
  (3) Forbidden word,
  (4) At least/less than certain frequency of letters.
- Length:
  (1) At least/less than certain number of words,
  (2) At least/less than certain number of sentences,
  (3) Exact number of paragraphs.
- Detectable Content:
  (1) postscript,
  (2) Exact number of placeholders.
- Detectable Format:
  (1) Number of bullet points,
  (2) Add title,
  (3) Answer from options,
  (4) Minimum of highlighted sections,
  (5) Json format.
- Change Cases:
  (1) All uppercase,
  (2) All lowercase,
  (3) At least/less than certain number of all-capital words.
- Startend:
  (1) End the text with a certain sentence,
  (2) Wrap whole response in double quotation.
- Punctuation:
  (1) No commas in response.
- Language:
  (1) Respond with certain language.

Table 6: Instruction Examples (6 levels)

| |
| --- |
| Instruction (level 1): |
| Write something with a topic of film or tv or video. At the end of your response, please explicitly add a postscript starting with P.S. |
| Constraint Type: |
| (1) Detectable Content - postscript |
| Instruction (level 2): |
| Produce a brief writing piece with emphasis on culture. The answer should be in all lowercase letters, with no capitalization. Response must also contain exactly 3 bullet points in markdown format. Use * to indicate bullets, like: 
 * xyz. 
 * abc. 
 * opq. |

Table 6: Instruction Examples (continued)

| Constraint Type: |
| --- |
| (1) Change Cases - All lowercase |
| (2) Detectable Format - Number of bullet points |

Instruction (level 3):
Produce a text with a focus on food. Your answer must have a title contained in double angular brackets, such as «title». Make sure you don't use any commas. Make sure to include the words 'ingredients'.

| Constraint Type: |
| --- |
| (1) Detectable Format - Add title |
| (2) Punctuation - No commas in response |
| (3) Keywords - Include keyword |

Instruction (level 4):
Write a text with a technology theme. be sure the letter 'l' appears at least 7 times in your response. Your entire response should be in all lowercase letters (no capital letters whatsoever). The word "artificial" should not appear in your response. make sure the response has less than 82 words.

| Constraint Type: |
| --- |
| (1) Keywords - At least/less than certain frequency of letters |
| (2) Change Cases - All lowercase |
| (3) Keywords - Forbidden word |
| (4) Length: At least/less than certain number of words |

Instruction (level 5):
Assist me in writing on a scientific topic. Highlight at least 3 text sections, i.e. *highlighted section*. Mention the word "research" for less than 4 times. make sure that words with all capital letters appear less than 2 times. The number of sentences in your response should be less than 5. Include a title wrapped in double angular brackets, i.e. «title».

| Constraint Type: |
| --- |
| (1) Detectable Format - Minimum of highlighted sections |
| (2) Keywords - Include keyword at least/less than certain frequency |
| (3) Keywords - At least/less than certain frequency of letters |
| (4) Length: At least/less than certain number of sentences |
| (5) Detectable Format - Add title |

Instruction (level 6):
Write a short description of entrepreneurs. The very end of your entire response should read exactly like: Is there anything else I can help with? The total number of words in your response should be at least 23. Include a title wrapped in double angular brackets, i.e. «title». Separate your response into 2 parts, where each part is separated with ***. Your answer must contain exactly 4 bullet point in Markdown using the following format:
* Bullet point one.
* Bullet point two.
...
Don't forget to include the keywords risk-taking.

Table 6: Instruction Examples (continued)

Constraint Type:
(1) Startend - End the text with a certain sentence
(2) Length: At least/less than certain number of words
(3) Detectable Format - Add title
(4) Length: Exact number of paragraphs
(5) Detectable Format - Number of bullet points
(6) Keywords: Include keyword

## A.4 DETAILED EXPERIMENTS

We can observe that our methods consistently outperform baselines on differen LLMs.

Table 7: Performance of methods across levels 1 to 6 (Llama-3-8B-Instruct)

| Method | Level 1 | Level 2 | Level 3 | Level 4 | Level 5 | Level 6 |
|---|---|---|---|---|---|---|
| Vanilla | 89.1 | 71.7 | 56.4 | 44.2 | 30.4 | 20.4 |
| Reflxion | 88.8 | 72.1 | 57.5 | 41.5 | 30.0 | 20.9 |
| BSM | 89.2 | 71.9 | 56.0 | 41.3 | 28.8 | 19.5 |
| U-SC | 89.5 | 71.8 | 56.7 | 45.1 | 31.2 | 20.6 |
| Rejection Sampling | 90.8 | 80.9 | 64.5 | 52.9 | 39.8 | 31.0 |
| ReAct | 93.6 | 81.3 | 68.8 | 54.4 | 39.1 | 30.7 |
| CRITIC | 94.1 | 85.8 | 74.4 | 61.2 | 51.1 | 41.5 |
| DVR(coldstart) | 95.0 | 86.7 | 76.9 | 65.3 | 53.7 | 43.8 |
| DVR(warmstart) | 95.4 | 87.6 | 77.0 | 67.4 | 55.4 | 46.9 |

Table 8: Performance of methods across levels 1 to 6 (Llama-3.1-70B-Instruct-AWQ-INT4)

| Method | Level 1 | Level 2 | Level 3 | Level 4 | Level 5 | Level 6 |
|---|---|---|---|---|---|---|
| Vanilla | 95.5 | 83.7 | 72.4 | 63.2 | 51.3 | 35.9 |
| Reflexion | 95.3 | 83.5 | 72.8 | 63.0 | 51.6 | 36.1 |
| BSM | 95.0 | 84.5 | 72.6 | 64.8 | 49.6 | 34.2 |
| U-SC | 96.0 | 83.3 | 71.8 | 64.0 | 52.5 | 36.3 |
| Rejection Sampling | 97.3 | 90.8 | 83.2 | 72.2 | 63.8 | 50.7 |
| ReAct | 97.5 | 91.0 | 83.5 | 72.4 | 65.0 | 52.1 |
| CRITIC | 98.1 | 93.2 | 87.6 | 79.1 | 73.7 | 61.3 |
| DVR(coldstart) | 98.0 | 94.3 | 88.2 | 82.0 | 75.7 | 63.1 |
| DVR(warmstart) | 98.2 | 94.6 | 88.7 | 82.2 | 76.0 | 64.2 |

Table 9: Comparison Across Different Constraints Types (coldstart)

| Constraint Type | Mistral-7B | | Llama3-8B | | Llama3.1-8B | | Llama 3.1-70B | |
|---|---|---|---|---|---|---|---|---|
| | Vanilla | DVR | Vanilla | DVR | Vanilla | DVR | Vanilla | DVR |
| Detectable Content | 76.36 | 88.49 | 84.18 | 95.82 | 86.29 | 96.19 | 97.06 | 98.14 |
| Keywords | 76.04 | 84.32 | 83.84 | 87.53 | 84.94 | 88.77 | 87.88 | 92.05 |
| Punctuation | 24.34 | 71.31 | 91.03 | 95.47 | 97.01 | 98.29 | 98.38 | 98.38 |
| Case Change | 70.08 | 80.15 | 81.28 | 93.20 | 82.97 | 89.96 | 80.23 | 96.24 |
| Start End | 81.29 | 88.71 | 84.88 | 88.71 | 84.07 | 91.78 | 89.37 | 95.94 |
| Detectable Format | 69.59 | 81.30 | 81.57 | 89.29 | 84.69 | 92.38 | 90.52 | 95.56 |
| Language | 69.11 | 82.72 | 77.06 | 86.24 | 81.96 | 89.30 | 90.83 | 94.34 |
| Length Constraints | 50.42 | 72.61 | 65.29 | 79.66 | 68.55 | 83.93 | 80.50 | 90.17 |

## A.5 LLM SELF-VERIFY ABILITY

We evaluate LLMs on their ability to verify whether the responses meet the given constraints. As shown in Table 10, LLMs often fail to verify their outputs accurately, suggesting they lack the capacity to provide reliable feedback or reflection on their own responses.

Table 10: LLMs Self-verification Accuracy (%)

| Model | Mistral-7B | Llama3-8B | Llama3.1-8B |
|---|---|---|---|
| | 53.1 | 56.8 | 55.7 |

## A.6 FLUENCY AND READABILITY

Table 11: Descriptive Statistics of Responses (ComplexInstruct), where Coherence_or1 is first order coherence and Coherence_2 is the second order coherence.

| Model | Method | Readability ↑ | Perplexity ↓ | Coherence_or1 ↑ | Coherence_or2 ↑ |
|---|---|---|---|---|---|
| mistral7B | Vanilla | 62.24 | 18.22 | 0.61 | 0.59 |
| | DVR | 61.93 | 18.95 | 0.59 | 0.57 |
| llama3-8B | Vanilla | 63.77 | 18.49 | 0.57 | 0.57 |
| | DVR | 63.58 | 18.08 | 0.59 | 0.56 |
| llama3.1-8B | Vanilla | 63.43 | 19.68 | 0.62 | 0.61 |
| | DVR | 62.75 | 18.30 | 0.62 | 0.60 |
| llama3.1-70B | Vanilla | 61.96 | 17.99 | 0.64 | 0.63 |
| | DVR | 63.51 | 18.04 | 0.63 | 0.62 |

Table 12: Descriptive Statistics of Responses (CoDI)

| Model | Method | Readability ↑ | Perplexity ↓ | Coherence_or1 ↑ | Coherence_or2 ↑ |
|---|---|---|---|---|---|
| mistral7B | Vanilla | 63.62 | 15.27 | 0.82 | 0.81 |
| | DVR | 63.53 | 15.98 | 0.83 | 0.82 |
| llama3-8B | Vanilla | 63.79 | 14.25 | 0.79 | 0.76 |
| | DVR | 64.14 | 16.07 | 0.80 | 0.77 |
| llama3.1-8B | Vanilla | 62.18 | 16.27 | 0.81 | 0.83 |
| | DVR | 62.02 | 17.58 | 0.81 | 0.80 |

## A.7 EXPERIMENTS ON IFEVAL

We conduct experiments on IFEval (Zhou et al., 2023) which is an instruction-following benchmark widely used for industry. The IFEval dataset evaluates the instruction-following ability and is one of the core benchmarks used in the Open LLM Leaderboard (Hugging Face). We conduct experiments on Mistral-7B-v0.3 and the results are shown in Table 13. DVR outperforms all other baselines on IFEval benchmark.

Table 13: ISR (%) for IFEval Dataset

| Method | Vanilla | Reflexion | BSM | U-SC | Rejection Sample | ReAct | CRITIC | DVR |
|---|---|---|---|---|---|---|---|---|
| ISR | 47.32 | 47.13 | 47.87 | 46.95 | 53.23 | 53.97 | 55.53 | 60.44 |

## A.8 EXPERIMENTS ON GPT4-TURBO

We conduct experiments on GPT-4-turbo. Shown in Table 14, we can observe that GPT-4-turbo performs better than open-source models (Mistral and Llama). Surprisingly, applied on Llama3.1-

8B, DVR can still outperform GPT-4-turbo, indicating that DVR exploits the potential of the open-source model.

Table 14: Performance Comparison to GPT4-turbo

| Model | Level 1 | Level 2 | Level 3 | Level 4 | Level 5 | Level 6 |
|---|---|---|---|---|---|---|
| Mistral-7B | 77.0 | 55.3 | 34.1 | 19.9 | 12.4 | 6.3 |
| DVR (Mistral-7B) | 95.0 | 81.3 | 66.6 | 51.4 | 36.4 | 23.4 |
| Llama3.1-8B | 90.5 | 76.6 | 62.5 | 50.1 | 35.6 | 25.3 |
| DVR (Llama3.1-8B) | 95.2 | 88.7 | 79.2 | 69.7 | 60.5 | 49.6 |
| GPT-4-turbo | 95.3 | 88.4 | 78.8 | 65.2 | 53.7 | 42.6 |

## A.9 ROBUSTNESS OF DVR

We also conduct experiments to assess DVR's performance in the presence of tool errors. Two types of errors are introduced: random noise and systematic bias. Specifically, we evaluate the framework on instructions with length constraints, using 600 samples (from ComplexInstruct) for word count control and another 600 for sentence count control. A constraint example: "The response needs to be less than (or at least) x number of words/sentences." where x ranges from 10 to 100 for words and 3 to 5 for sentences. We add two types of noises to tools:

**Noise:** Gaussian noise with a mean of 0 is added to the counted number of words (or sentences) to simulate random errors. The DVR's performance is then measured across different deviation levels.

**Bias Errors:** A fixed bias is added to the counted values of words (or sentences) to introduce systematic errors. The tables below demonstrate DVR's performance under different bias values.

**Observations:** We have several observations in Table 15 and Table 16. (1) The performance will decrease as the noise levels (deviation, bias values) increase. (2) As the errors become large, the performance degradation will saturate. (3) Overall, DVR will not perform much worse than vanilla even if the bias and errors are large (20 for word count and 4 for sentence count). (4) The impact of noise on the overall instruction satisfaction rate is less severe compared to its influence on specific constraints.

Table 15: Satisfaction Rate for Word Number Constraints (%)

| Deviation | 0 | 5 | 10 | 20 | Vanilla |
|---|---|---|---|---|---|
| Words Number Satisfaction Rate | 88.00 | 87.17 | 82.83 | 81.50 | 68.16 |
| Instruction Satisfaction Rate | 48.17 | 45.00 | 43.67 | 43.67 | 10.17 |

| Bias | 0 | 5 | 10 | 20 | Vanilla |
|---|---|---|---|---|---|
| Words Number Satisfaction Rate | 88.00 | 87.17 | 84.33 | 82.67 | 68.16 |
| Instruction Satisfaction Rate | 48.17 | 47.83 | 47.00 | 45.67 | 10.17 |

Table 16: Satisfaction Rate for Sentence Number Constraints (%)

| Deviation | 0 | 1 | 2 | 4 | Vanilla |
|---|---|---|---|---|---|
| Sentences Number Satisfaction Rate | 74.50 | 68.17 | 62.50 | 60.50 | 56.33 |
| Instruction Satisfaction Rate | 42.83 | 38.17 | 35.33 | 34.33 | 10.17 |

| Bias | 0 | 1 | 2 | 4 | Vanilla |
|---|---|---|---|---|---|
| Sentences Number Satisfaction Rate | 74.50 | 64.67 | 58.67 | 56.50 | 56.33 |
| Instruction Satisfaction Rate | 42.83 | 40.50 | 32.67 | 31.33 | 10.17 |

## A.10 COMPUTATION TIME

We conducted experiments with 20 instructions, each containing 6 constraints, using Mistral-7B. The number of trials was set to 5, consistent with the paper's settings. The average running time is summarized below:

Table 17: The Average Running Time for One Sample

| Method | Vanilla | Reflexion | U-SC | BSM | Rejection Sample | ReAct | CRITIC | DVR |
|---|---|---|---|---|---|---|---|---|
| Running Time (s) | 5.91 | 20.53 | 41.34 | 46.21 | 32.32 | 36.48 | 37.98 | 33.91 |

As shown in Table 17, our method does not exhibit significantly higher running time compared to other baselines. Considering the performance gains (Table 2), our method demonstrates a balance between efficiency and effectiveness.

### A.11 LIMITATIONS AND FUTURE WORKS

There are several limitations and potential future works.

(1) Currently, we consider multiple independent constraints. However, the instructions in real-world might be more complex and constraints might have dependency with each other (Wen et al., 2024). For example, the instruction can ask the response has 4 bullet points and 2 sentences in each bullet point. In such scenario, simple decomposition might not work. (2) Moreover, tools may not be available for new constraints. Here, we assume that we have tools for all existing constraints. However, users' requirements can be very diverse and we would not have certain tools for new constraints. (3) As shown in Table 15 and Table 16, DVR performance declines when tools produce errors.

As a result, we can consider the following future works. (1) The first one is to handle constraints with dependency. If LLMs can correctly disentangle these constraints and find appropriate tools, our framework can also enhance LLMs' instruction-following ability in such a complex setting. (2) The second future work is to deal with new constraints. One possible solution is to firstly find out if the constraint can be tested via python code. Then local LLMs can summarize constraints into code requirements and send them to advanced code generators for tool generation. Generated tools can also be saved locally for future use. (3) Ensuring the reliability and quality of tools is crucial for real-world applications. There are techniques such as developing test cases that verify a tool's outputs against known values (Cai et al., 2024; Huang et al., 2023) or generating multiple versions of a tool and selecting the most consistent output. Additional research is needed to evaluate whether these techniques are sufficient to guarantee tool quality. Furthermore, new approaches could be explored to enhance tool generation.

### A.12 CASE STUDY

We use some examples to show how LLMs modify responses with the guidance of feedback. The red colored content is the errors from the original response and blue colored content is the corrected parts after refinement.

Table 18: Refinement Examples

| |
|---|
| Prompt: |
| Provide me with text related to the everyday experience. make sure the response has less than 80 words. Respond with at least 3 sentences. The entire reply should contain a title in double angular brackets, i.e. «title». Your entire response should be in all lowercase letters (no capital letters whatsoever). |
| Response: |
| «morning routines» |
| i love starting my day with a quiet cup of coffee and a good book. it helps me set a positive tone for the rest of the day. morning routines can be as simple or as elaborate as you like, but they help create a sense of structure and calm in our otherwise chaotic lives. [name]'s morning routine involves a quick workout followed by a healthy breakfast. [another name] prefers to meditate for a few minutes before beginning their day. |
| Unsatisfied Constraint: make sure the response has less than 80 words. |

Table 18: Refinement Examples (continued)

| |
|---|
| Feedback: 
 The response contains 83 words, which is 4 too many. 
 Please remove 4 words to have fewer than 80 words. |
| Refined Response: «morning routines» 
 i love starting my day with a quiet cup of coffee and a good book. it helps me set a positive tone. morning routines can be as simple or as elaborate as you like, but they help create a sense of structure and calm in our chaotic lives. [name]'s morning routine involves a quick workout followed by a healthy breakfast. |

| |
|---|
| Instruction: 
 Excuse me, could you please provide text that focuses on the music-topic? use the keywords artists. highlight at least 1 key point by wrapping it with *. For example: *highlighted key point*. The number of sentences in your response should be less than 5. be sure the letter e appears at least 4 times in your response. Make sure to include at least 3 placeholder represented by square brackets, such as [address], [name]. the total number of words in your response should be at least 23. |
| Response: 
 «Music Unites» From festivals to online communities, music continues to bridge cultural and geographical gaps like never before. *Electronic music has especially seen a surge in popularity among the younger generation*. Talented artists from all over the world have been coming together to create beautiful melodies and harmonies. [name] is one such artist who has been making waves in the industry with his eclectic beats. His music can be found on various platforms, including [address] and [website]. Many emerging artists, including [another name], are also experimenting with different genres. |
| Unsatisfied Constraint: The number of sentences in your response should be less than 5. |
| Feedback: 
 The response contains 6 sentences, which is 2 more than allowed. 
 Please remove at least 2 sentences. |
| Refined Response: 
 «Music Unites» From festivals to online communities, music continues to bridge cultural and geographical gaps like never before. *Electronic music has especially seen a surge in popularity among the younger generation*. Talented artists from all over the world have been coming together to create beautiful melodies and harmonies. [name] is one such artist who has been making waves in the industry with his eclectic beats and his music can be found on various platforms, including [address] and [website]. |

| |
|---|
| Instruction: 
 Kindly generate a few texts on the subject of education or learning. Give a title wrapped in double angular brackets, such as «your title». Add stress words which are capitalized. Ensure those stress words appear less than 4 times. answer with less than 99 words. The very last sentence of your response should be "Hope you agree with me.". |
| Response: 
 «The Power of Learning» 
 LEARNING IS A LIFELONG PROCESS that enables individuals to acquire new skills and knowledge. It plays a CRUCIAL role in shaping one's career and personal growth. Education helps to broaden one's perspective and fosters critical thinking. Hope you agree with me. |
| Unsatisfied Constraint: Add stress words which are capitalized. Ensure those stress words appear less than 4 times. |
| Feedback: The response contains 6 capitalized words, which is more than the allowed 3. 
 Please remove at least 3 capitalized word(s). |

Table 18: Refinement Examples (continued)

---

Refined Response:

«The Power of Learning»

Learning is a lifelong process that enables individuals to acquire new skills and knowledge. It plays a crucial role in shaping one's career and personal growth. Education is VERY important and helps to broaden one's perspective. Hope you agree with me.

---

Instruction: Produce a text about TV. Your entire response should be in all capital letters. Your response should contain at least 3 sentences. Your answer must have a title contained in double angular brackets, such as «title». Make sure to include at least 3 placeholder represented by square brackets, such as [address], [name]. Highlight at least 2 sections of your response in markdown such as *highlighted section*.

---

Response:

«TV SHOWS FOR THE AGES»

TV SHOWS HAVE BEEN A STAPLE OF ENTERTAINMENT FOR DECADES. FROM CLASSICS LIKE Ï LOVE LUCYÏTO MODERN HITS LIKE ĠAME OF THRONESÏHERE IS SOMETHING FOR EVERYONE. *TV PROVIDES A WAY FOR US TO ESCAPE INTO DIFFERENT WORLDS AND EXPERIENCE STORIES THAT WE MIGHT NOT OTHER-WISE ENCOUNTER*. WITH THE RISE OF STREAMING SERVICES LIKE NETFLIX AND HULU IT IS EASIER THAN EVER TO ACCESS YOUR FAVORITE SHOWS FROM THE COMFORT OF YOUR OWN HOME AT [address]. *WHETHER YOU ARE A FAN OF DRAMA COMEDY OR REALITY TV THERE IS NO SHORTAGE OF OPTIONS TO CHOOSE FROM*. FANS OF [name] CAN LOOK FORWARD TO NEW CONTENT FROM THEIR FAVORITE CREATORS AND DISCOVER NEW SHOWS FROM TALENTED UP-AND-COMING WRITERS AND PRODUCERS. WITH SO MANY GREAT SHOWS TO CHOOSE FROM IT IS NO WONDER THAT TV REMAINS A POPULAR FORM OF EN-TERTAINMENT FOR PEOPLE ALL OVER THE WORLD. [channel] OFFERS A WIDE VARIETY OF PROGRAMMING THAT CATERS TO DIFFERENT TASTES AND PREF-ERENCES.

---

Unsatisfied Constraint: Your entire response should be in all capital letters.

Feedback: The response contains words not in all capital letters: [address], [name], [channel]. Please capitalize all of them.

Refined Response:

«TV SHOWS FOR THE AGES»

TV SHOWS HAVE BEEN A STAPLE OF ENTERTAINMENT FOR DECADES. FROM CLASSICS LIKE Ï LOVE LUCYÏTO MODERN HITS LIKE ĠAME OF THRONESÏHERE IS SOMETHING FOR EVERYONE. *TV PROVIDES A WAY FOR US TO ESCAPE INTO DIFFERENT WORLDS AND EXPERIENCE STORIES THAT WE MIGHT NOT OTH-ERWISE ENCOUNTER*. WITH THE RISE OF STREAMING SERVICES LIKE NET-FLIX AND HULU IT IS EASIER THAN EVER TO ACCESS YOUR FAVORITE SHOWS FROM THE COMFORT OF YOUR OWN HOME AT [ADDRESS]. *WHETHER YOU ARE A FAN OF DRAMA COMEDY OR REALITY TV THERE IS NO SHORTAGE OF OP-TIONS TO CHOOSE FROM*. FANS OF [NAME] CAN LOOK FORWARD TO NEW CONTENT FROM THEIR FAVORITE CREATORS AND DISCOVER NEW SHOWS FROM TALENTED UP-AND-COMING WRITERS AND PRODUCERS. WITH SO MANY GREAT SHOWS TO CHOOSE FROM IT IS NO WONDER THAT TV REMAINS A POPULAR FORM OF ENTERTAINMENT FOR PEOPLE ALL OVER THE WORLD. [CHANNEL] OF-FERS A WIDE VARIETY OF PROGRAMMING THAT CATERS TO DIFFERENT TASTES AND PREFERENCES.

---

Table 18: Refinement Examples (continued)

| |
|---|
| **Instruction:** Generate a writing piece related to health. organize your entire response in at least 3 sentences. Answer should be written with all letters lowercased. In your entire response, the letter n should appear less than 6 times. |
| **Response:** maintaining a healthy lifestyle involves making conscious choices about the food we eat and the activities we engage in. eating a balanced diet that includes plenty of fruits and vegetables can help to boost our energy levels and support overall well-being. regular exercise, such as walking or yoga, can also have a positive impact on both our physical and mental health. |
| **Unsatisfied Constraint:** In your entire response, the letter n should appear less than 6 times. |
| **Feedback:** The response contains the letter 'n' 23 times, which is 18 too many. Please remove 18 occurrences to have less than 6. |
| **Refined Response:** eating a healthy diet is good for us. it helps to keep us strong. we should try to do it. |

| |
|---|
| **Instruction:** Oh, dear one, would you be so kind as to grace me with a magnificent text delving into the ever-so-mysterious world of science? The very end of your entire response should read exactly like: Let me know if you have additional questions. All letters in your response must be lower case letters. Separate your response into 4 parts, where each part is separated with ***. use the keywords physics. Highlight at least 3 text sections, i.e. *highlighted section*. there should be at least 4 occurrences of the word ïnnovationsïn your response. |
| **Response:** «exploring the mysteries of science» 

 *** *the wonders of physics* have long been a source of fascination for scientists and the general public alike. from the behavior of subatomic particles to the vast expanses of space, physics governs the fundamental laws of our universe. 

 *** *innovations in physics* have led to numerous breakthroughs in fields such as medicine, technology, and engineering. for instance, advances in medical imaging have allowed for more accurate diagnoses and targeted treatments. 

 *** *the intersection of physics and technology* has given rise to exciting innovations such as quantum computing, nanotechnology, and renewable energy solutions. these innovations have the potential to transform industries and improve lives worldwide. 

 *** as scientists continue to push the boundaries of human knowledge, we can expect to see even more groundbreaking innovations in the field of physics. let me know if you have additional questions. |
| **Unsatisfied Constraint:** Separate your response into 4 parts, where each part is separated with ***. |
| **Feedback:** Expected exactly 4 sections, but found 5. Please remove 1 section by removing 1 '***'. |
| **Refined Response:** «exploring the mysteries of science» 
 the wonders of physics* have long been a source of fascination for scientists and the general public alike. from the behavior of subatomic particles to the vast expanses of space, physics governs the fundamental laws of our universe. 

 *** 
 innovations in physics* have led to numerous breakthroughs in fields such as medicine, technology, and engineering. for instance, advances in medical imaging have allowed for more accurate diagnoses and targeted treatments. |

Table 18: Refinement Examples (continued)

---

***

the intersection of physics and technology* has given rise to exciting *innovations* such as quantum computing, nanotechnology, and renewable energy solutions. these *innovations* have the potential to transform industries and improve lives worldwide. and these *innovations* are just the beginning.

*** as scientists continue to push the boundaries of human knowledge, we can expect to see even more groundbreaking *innovations* in the field of physics. let me know if you have additional questions.

---

Instruction: Provide me with a text that revolves around the theme of family. do not say the word "relationships" in the response. Highlight at least 2 sections of your response in markdown such as *highlighted section*. Make sure to include a postscript starting with P.S. Have at least 5 sentences in your response. make sure the response has less than 56 words. Your answer must contain exactly 2 bullet points in the markdown format (use * to indicate each bullet) such as:
* This is the first point.
* This is the second point.

Response: A family is a bond that can never be broken. *Children grow up with love and support from their family*. As they grow older they begin to appreciate the sacrifices their family has made for them.
* A family can provide a sense of belonging.
* A family can provide a sense of security.
P.S. I love my family.

---

Unsatisfied Constraint: Highlight at least 2 sections of your response in markdown such as *highlighted section*.

Feedback: The response contains 1 highlighted section.
Please include 1 more highlighted section in the format of *highlighted*.

---

Modified Response: A family is a *bond that can never be broken*. *Children grow up with love and support from their family*. As they grow older they begin to appreciate the sacrifices their family has made for them.
* A family can provide a sense of belonging.
* A family can provide a sense of security.
P.S. I love my family.

---

Instruction: Produce a comprehensive written discourse analyzing the multiple facets of film. Answer in lowercase letters only, throughout your entire answer. There should be exactly 4 paragraphs in your response, separated by the markdown divider: ***. Response must also contain exactly 3 bullet points in markdown format. Use * to indicate bullets, like:
* xyz
* abc
* opq

---

Response:

Table 18: Refinement Examples (continued)

«the art of storytelling in film»

storytelling is a fundamental aspect of filmmaking. it is the art of conveying emotions and ideas through visual narratives. films have the power to transport us to different worlds, evoke emotions, and make us think. whether it's a romantic comedy or a sci-fi epic, storytelling is the core of what makes a film engaging.
***
film is a collaborative medium that requires the involvement of numerous individuals, from screenwriters to cinematographers. the director plays a crucial role in bringing the story to life, guiding the actors and crew to create a cohesive vision. the cinematographer's job is to capture the visuals that will transport the audience to the world of the story.
***
* some notable film movements include:
* german expressionism
* french new wave
* italian neorealism
each of these movements brought a unique perspective to filmmaking, experimenting with style and narrative structure. they paved the way for future generations of filmmakers, allowing them to push the boundaries of what is possible on screen.
***
ultimately, film is a powerful medium that can inspire, educate, and entertain. it has the ability to transcend cultural boundaries, allowing us to connect with people from different backgrounds and experiences. whether you're a filmmaker or a film enthusiast, the world of cinema has something to offer everyone.

---

Unsatisfied Constraint: Response must also contain exactly 3 bullet points in markdown format. Use * to indicate bullets, like:
* xyz
* abc
* opq

---

Feedback:
In the response, there are 4 bullet points.
Here are the bullet points detected:
* some notable film movements include:...
* german expressionism
* french new wave
* italian neorealism
Please remove exactly 1 bullet points to meet the requirement of 3.

Table 18: Refinement Examples (continued)

Refined Response:
«the art of storytelling in film»
storytelling is a fundamental aspect of filmmaking. it is the art of conveying emotions and ideas through visual narratives. films have the power to transport us to different worlds, evoke emotions, and make us think. whether it's a romantic comedy or a sci-fi epic, storytelling is the core of what makes a film engaging.
***
film is a collaborative medium that requires the involvement of numerous individuals, from screenwriters to cinematographers. the director plays a crucial role in bringing the story to life, guiding the actors and crew to create a cohesive vision. the cinematographer's job is to capture the visuals that will transport the audience to the world of the story.
***
* some notable film movements include:
* french new wave
* italian neorealism
each of these movements brought a unique perspective to filmmaking, experimenting with style and narrative structure. they paved the way for future generations of filmmakers, allowing them to push the boundaries of what is possible on screen.
***
ultimately, film is a powerful medium that can inspire, educate, and entertain. it has the ability to transcend cultural boundaries, allowing us to connect with people from different backgrounds and experiences. whether you're a filmmaker or a film enthusiast, the world of cinema has something to offer everyone.

Table 19: Refinement Examples on CoDI dataset

Instruction: Generate a text that involves anger and is related to films:

Response: The director's blatant disregard for the script and the talent on set is nothing short of astounding. They have the audacity to claim artistic vision while disregarding the very actors they hired to bring their mediocre ideas to life. It's a travesty that someone so clearly out of touch can call themselves a filmmaker. The industry should be ashamed for allowing such incompetence to flourish.

Feedback: The sentiment of the text is 'disgust', which does not match the required sentiment 'anger'.
Please adjust the sentiment of the text to be more 'anger'.

Refined Response:
The director's flagrant disregard for the script and the talent on set is infuriating. They have the nerve to claim artistic vision while callously disregarding the very actors they hired to bring their subpar ideas to life. It's outrageous that someone so clearly out of touch can call themselves a filmmaker. The industry should be held accountable for allowing such incompetence to perpetuate. It's not just a minor mistake; it's a systemic issue that demands attention. The fact that this director continues to receive opportunities is a slap in the face to every talented artist who has been overlooked and underappreciated. The situation is unacceptable, and it's time for change.
The final answer is: There is no one final answer to this problem as it involves modifying the sentiment of multiple pieces of writing to fit the constraints provided. The provided code and response modifications demonstrate how to adjust the sentiment to match the required tone for each instruction.

Instruction: Generate text on science with disgust, using general words:

Table 19: Refinement Examples on CoDI (continued)

Response: The discoveries in the realm of biology have been grotesque, with the revelation of creatures that defy natural explanation. Their twisted forms and behaviors evoke a sense of revulsion, as if the very fabric of life has been warped in ways that are both fascinating and repulsive. It's a field that challenges our understanding of what it means to be alive.

Feedback: The detected topic of the response is learning and education, which does not match the expected topic science and technology.

Please adjust the content to align more closely with the topic science and technology.

Refined Response:
The latest breakthroughs in genetic engineering have been abhorrent, with the development of organisms that are unnatural and unsettling. Their abnormal structures and functions elicit a strong sense of disgust, as if the boundaries of science have been overstepped. It's a field that tests our understanding of the fundamental principles of life and the ethics of scientific inquiry.

## A.13 PROMPTS

Table 20: Prompts

| Prompt for Response Generation |
| --- |
| Generation prompt: |
| You are an AI assistant that generates responses based on given prompts. |
| For each prompt, provide a response that adheres to the specified constraints. |
| #Prompt: Generate a text that touch on arts. make sure the response has less than 40 words. make sure it contains at least 2 placeholders represented by square brackets, such as [name]. Include a title wrapped in double angular brackets, i.e. «title». |
| Response: «The Beauty of Expression» |
| Art, from [name]'s perspective, transforms emotions into visual narratives. [another name]'s masterpieces illustrate this beautifully. |
| #Prompt: Generate sth. about celebrity or pop culture. the response must contain a title wrapped in double angular brackets, i.e. «title». The answer should be in all lowercase letters, with no capitalizations. The word ëntertainments̈hould not appear in your response. In your entire response, the letter m should appear at least 3 times. Your answer must have at least 2 placeholders, wrapped in square brackets, such as [author]. |
| ...(more examples) |
| #Prompt: {current instruction} |
| Response: |

| Prompt for Instruction Decomposition |
| --- |

Table 20: Prompts (continued)

You are an advanced assistant specializing in identifying and listing output constraints from provided instructions. The instructions typically include a task related to generating content on a specific topic and one (or multiple) format constraint(s). Your goal is to focus only on extracting and listing all the format constraints required for the output, ignoring the content-related task.

Instruction:

Please generate a few lines of text that touch on the topic of tv. Put your entire answer in JSON format. The word 'show' should not appear in your response. Use square brackets for placeholders, like [username1], [username2]. Please include at least 2 placeholders in the thread.You are not allowed to use any commas in your response.

Format Constraints:

#1. Put your entire answer in JSON format.

#2. The word 'show' should not appear in your response.

#3. Use square brackets for placeholders, like [username1], [username2]. Please include at least 2 placeholders in the thread.

#4. You are not allowed to use any commas in your response.

... (more examples)

Instruction:

{current instruction}

Format Constraints:

---

Prompt for Tool Selection

You will be given a list of constraints. Each constraint belongs to a specific category. Your task is to recognize and categorize each constraint. Only output the category from the following options:

postscript, placeholder, include keyword, exclude keyword, letter frequency, keyword frequency, sentence count constraint, word count constraint, *** separator, bullet points, fixed responses, highlighted, JSON format, title format, quoted response, end phrase, no commas, all capital letters, all lowercase, capital word frequency, language restriction

Please ensure to categorize each constraint accurately according to its description. There is definitely a valid category option for each constraint. You can Here are examples for each type of constraint:

Prompt: Make sure to include the word 'mutations'.

Category: include keyword

Prompt: Limit the number of words you use to fewer than 65 words.

Category: word count constraint

...(more examples)

Prompt: {Current Prompt}

Category:

---

Prompt for Refinement

Table 20: Prompts (continued)

You are an AI assistant responsible for refining a given response. Given a prompt, its original response, and the analysis of the response, your task is to modify the response according to the analysis.

#Prompt: I'm looking for text that explores arts or culture, can you assist? please explicitly add a note starting with P.S. There should be exactly 2 paragraphs in your response, separated by the markdown divider: ***. Make sure to include at least 2 placeholder represented by square brackets, such as [address], [name]. Highlight at least 2 text sections, i.e. *highlighted section*. There should be no commas in your reply.

#Original Response: Art has the power to bring people together and transcend cultural boundaries. It can evoke emotions and spark conversations that might not be possible through other means. *At the [address] museum, visitors can experience this firsthand by exploring the diverse collection of art from around the world.*

*** From paintings to sculptures to installations, each piece tells a unique story that can be interpreted in many ways. *The work of [name] is a great example of this, as it challenges viewers to think critically about the world around them.* Whether you're an art enthusiast or just looking for a new perspective, the [address] museum is a must-visit destination. P.S. Don't forget to check out the museum's events calendar for upcoming exhibitions and performances!

#It does not satisfy the constraint: There should be no commas in your reply.

#Analysis: The response contains 4 comma(s). Here are the detected commas: ( museum, visitors) (tallations, each ) ( of this, as it ) (perspective, the [address).
Please remove all commas.

#Modified Response: Art has the power to bring people together and transcend cultural boundaries. It can evoke emotions and spark conversations that might not be possible through other means. *At the [address] museum visitors can experience this firsthand by exploring the diverse collection of art from around the world.*

*** From paintings to sculptures to installations each piece tells a unique story that can be interpreted in many ways. *The work of [name] is a great example of this as it challenges viewers to think critically about the world around them.* Whether you're an art enthusiast or just looking for a new perspective the [address] museum is a must-visit destination. P.S. Don't forget to check out the museum's events calendar for upcoming exhibitions and performances!

....(more examples)

#Prompt: current prompt

#Original Response: current response

#It does not satisfy the constraint: current unsatisfied constraint

#Analysis: current feedback

#Modified Response:

## A.14    FAILURE CASES

We analyze these failure cases and find out two typical situations that our framework would fail:

**(1) Combined Effect:** In certain scenarios, multiple constraints can create a challenge for the LLMs to balance. As shown in the first example in the Table 21, in the case of limiting the frequency of the letter 't' while generating content about sports, the natural occurrence of the letter 't' in 'sports', combined with 2 sections requirements, makes it difficult to satisfy 't' letter frequency limit. Although these constraints are not inherently contradictory, their combined effect puts the model under pressure, resulting in failure to meet some requirements. **(2) Incorrect decomposition and wrong tool selection:** Incorrect decmoposition would lead to wrong tool selection. As shown in the second example in the Table 21, the constraint to add capitalized stress words fewer than four times was mistakenly decomposed into two separate tasks. This leads to the incorrect selection of tools, taking the "stress words" as keywords instead of capitalized words. This type of failure degrades the overall performance in following complex instructions.

Table 21: Failure Cases

Instruction: Generate text on the subject of sports. Make sure to highlight at least 2 sections in your answer with markdown, i.e. use *highlighted section*. Please also include the keywords "games" in the response. You are not allowed to use any commas in your response. Put the response into 2 sections, separated using 3 asterisks ***. Please make sure each point have a title wrapped in double angular brackets, i.e. «title». In your entire response, the letter t should appear less than 10 times.

«Sports Overview»
Participating in sports is an excellent way to improve physical and mental health *Sports like basketball and soccer require quick reflexes and agility* Playing sports also fosters teamwork and communication among players.
***
«The Thrill of Games»
Competitive games bring out the best in athletes *The rush of adrenaline during a crucial match is exhilarating* Fans enjoy the suspense and excitement of watching their favorite teams battle it out.

Unsatisfied Constraint: In your entire response, the letter t should appear less than 10 times.

Discussion:
There are other constraints against this one. The topic is about sports which contains the letter t. Additionally, "Put the response into 2 sections" tends to make the response long, wich makes satisfying letter frequency limit challenging.

Instruction: Can you provide me with some information about dining? Add stress words which are capitalized. Ensure those stress words appear less than 4 times.

Response:
«The Dining Experience»

A DELICIOUS meal is not just about the food, but also about the atmosphere and service. When dining out, it's essential to consider the AMBIANCE of the restaurant, as it can greatly impact your overall experience. A restaurant with a cozy and intimate setting can make for a ROMANTIC evening, while a lively and bustling atmosphere can be PERFECT for a night out with friends.

Furthermore, the service can make or break your dining experience. A friendly and attentive wait staff can make you feel welcome and valued, while poor service can leave a sour taste in your mouth.

Ultimately, a great dining experience is one that combines DELICIOUS food with exceptional service and a pleasant atmosphere.

Unsatisfied Constraint: Add stress words which are capitalized. Ensure those stress words appear less than 4 times.

Table 21: Failure Cases (continued)

Discussion:

In this example, the constraint "Add stress words which are capitalized. Ensure those stress words appear less than 4 times" is mistakenly decomposed into two separate constraints by the LLMs: (1) Add stress words which are capitalized. (2) Ensure those stress words appear less than 4 times. For the first constraint, the LLMs can not find a tool for it. For the second constraint, the LLM incorrectly selects the keyword tool: Keywords("stress words", "less than", 4). In this case, the correct tool should be Capitalwords("less than",4). Usually, incorrect decomposition prevents LLMs from identifying any tools, rather than causing them to select an incorrect one. This example is unique because the constraint is phrased in two sentences, and the second sentence can be misinterpreted to suggest a limit on keywords (i.e., "stress words") instead of focusing on capitalized stress words.

### A.15  TOOL EXAMPLES

**1. Word Counting:** This example is obtained through GPT4-o with zero-shot. It demonstrates that reliable tools can be easily created. The details are as follows:

```
def feedback(response, max_words=None, min_words=None):
    # Count the number of words in the response
    word_count = len(response.split())

    # Check for maximum word constraint
    if max_words is not None and word_count > max_words:
    return f"Response failed because it has {word_count} words,
    exceeding the maximum allowed limit of {max_words} words."

    # Check for minimum word constraint
    if min_words is not None and word_count < min_words:
    return f"Response failed because it has only {word_count}
    words, fewer than the minimum required {min_words} words."

    # If all constraints are satisfied
    return True
```

**Lowercase Letter Validation:** This example validates whether a given text is entirely in lowercase. If any word contains uppercase letters, it provides feedback on which words need correction. The implementation is as follows:

```
class LowercaseLetter:
    def __init__(self):
        pass

    def feed_back(self, value):
        # Split the input string into words
        words = value.split()

        # Find words that are not fully in lowercase
        upper_case_words = [word for word in words if
        any(char.isupper() for char in word)]

        if value.islower():
            return True
        else:
            return f"The response contains words that are not in
            all lowercase letters: {', '.join(upper_case_words)}.
            Please lowercase all of them."
```

