# OpenReview forum: "Divide-Verify-Refine: Aligning LLM Responses with Complex Instructions"
_ICLR.cc/2025/Conference — Submitted to ICLR 2025_

### Official Review · Reviewer_6oxA · 2024-10-30

**Soundness:** 2
**Presentation:** 3
**Contribution:** 2
**Rating:** 5
**Confidence:** 3

**Summary:**

The paper presents the Divide-Verify-Refine (DVR) framework, which enhances the ability of Large Language Models (LLMs) to follow complex instructions with multiple constraints. The DVR framework consists of three steps: dividing complex instructions into single constraints, verifying responses with external tools for reliable feedback, and refining responses using a repository of successful refinement examples. The framework is evaluated on two datasets, showing significant improvements in constraint adherence without the need for retraining. It addresses the challenges of feedback quality and constraint diversity by integrating tools and leveraging past experiences, respectively. The paper concludes that DVR offers a scalable solution to enhance the practical usability of LLMs in real-world applications.

**Strengths:**

- The DVR framework significantly enhances LLMs' ability to adhere to complex instructions containing multiple constraints, which is crucial for mission-critical applications.

- Unlike fine-tuning approaches, DVR improves LLM performance without the need for extensive retraining, making it more accessible and less computationally expensive.

- By leveraging external tools for verification, DVR provides a more reliable feedback mechanism than LLMs' self-reflection, which tends to be unreliable.

**Weaknesses:**

- When preparing appropriate tools for verification, how accurate are the tools? Are there many problems with the tools themselves? If there is a problem with the tools, it can cause problems in all subsequent modules.
- For completely new constraints, there may not be examples where a tool is readily available. Imagine a scenario where we need to develop a large language model (LLM) to generate text in a specific style, say "Gothic" style, but currently there is no tool to directly verify if the text is in this specific style.
- DVR has more flow modules, which results in the experimental setup not being very solid.

**Questions:**

No questions.

---

> ### Author Response · Authors · 2024-11-20
>
> Thanks for your comments and questions. We will answer them one by one.
>
> ### Tool Correctness
>
> 1. We appreciate the reviewer for highlighting the concern regarding the reliability of newly generated tools. (a) However, it is essential to clarify that ensuring the correctness of these tools falls outside the scope of our study, as it relates more closely to the code-generation field, which have been extensively studied such as [1,2,3,4]. The tool making and verification process are also studied in ICLR24 [1]. There are techniques such as developing test cases that verify a tool’s outputs against known values [1,2] or generating multiple versions of a tool and selecting the most consistent output. (b) The “tool” in our setting is not limited to codes. They can also be APIs that widely exist in the real world [5]. (c) The tools used to verify responses are overall simple and easy to develop (e.g., counting words code). Down below, we also show an example of a tool obtained from GPT-4o with zero-shot. These tools are easy to develop, but LLM struggles to do the same job by itself (e.g., counting words). (d) In many real-world scenarios, constraints are predefined, allowing high-quality tools to be pre-prepared. For instance, in legal or technical writing, tools for format validation or terminology checks can be readily established.
>
> [1] Cai, Tianle, et al. *"Large Language Models as Tool Makers."* ICLR 2024.
> [2] Huang, Dong, et al. *"Agentcoder: Multi-agent-based code generation with iterative testing and optimisation."* arXiv:2312.13010 (2023).
> [3] Zhang, Shun, et al. "Planning with Large Language Models for Code Generation." ICLR 2023.
> [4] Jiang, Xue, et al. "Self-planning code generation with large language models." ACM Transactions on Software Engineering and Methodology.
> [5] Qu, Changle, et al. *"Tool Learning with Large Language Models: A Survey."* arXiv:2405.17935 (2024).
>
> 2. **Influence of Potential Errors** Although improving the quality of tools is out of the scope of our paper, we also conduct experiments to assess DVR’s performance in the presence of tool errors. Two types of errors are introduced: random noise and systematic bias. Specifically, we evaluate the framework on instructions with length constraints, using 600 samples (from ComplexInstruct) for word count control and another 600 for sentence count control. A constraint example: “The response needs to be less than (or at least) x number of words/sentences.” where x ranges from 10 to 100 for words and 3 to 5 for sentences.  We add two types of noises to tools: **Noise**: Gaussian noise with a mean of 0 is added to the counted number of words (or sentences) to simulate random errors. The DVR’s performance is then measured across different deviation levels. **Bias Errors**: A fixed bias is added to the counted values of words (or sentences) to introduce systematic errors. The tables below demonstrate DVR’s performance under different bias values.
>
> **Observations:** We have several observations in Table 1 and Table 2. (1) The performance will decrease as the noise levels (deviation, bias values) increase. (2) As the errors become large, the performance degradation will saturate. (3) Overall, DVR will not perform much worse than vanilla even if the bias and errors are large (20 for word count and 4 for sentence count). (4) The impact of noise on the overall instruction satisfaction rate is less severe compared to its influence on specific constraints.
>
> **Table 1. Satisfaction Rate (%) for word number constraints**
>
> | Deviation | 0    | 5    | 10    | 20    | Vanilla |
> |-----------|------|------|-------|-------|---------|
> | Words Number Satisfaction Rate | 88.00 | 87.17 | 82.83 | 81.50 | 68.16   |
> | Instruction Satisfaction Rate  | 48.17 | 45.00 | 43.67 | 43.67 | 10.17   |
>
> | Bias      | 0    | 5    | 10    | 20    | Vanilla |
> |-----------|------|------|-------|-------|---------|
> | Words Number Satisfaction Rate | 88.00 | 87.17 | 84.33 | 82.67 | 68.16   |
> | Instruction Satisfaction Rate| 48.17 | 47.83 | 47.00 | 45.67 | 10.17   |
>
> ---
>
> **Table 2. Satisfaction Rate for Sentence Number Constraints**
>
>
> | Deviation                          | 0     | 1     | 2     | 4     | Vanilla |
> |------------------------------------|-------|-------|-------|-------|---------|
> | Sentences Number Satisfaction Rate | 74.50 | 68.17 | 62.50 | 60.50 | 56.33   |
> | Instruction Satisfaction Rate | 42.83 | 38.17 | 35.33 | 34.33 | 10.17   |
>
> | Bias                               | 0     | 1     | 2     | 4     | Vanilla |
> |------------------------------------|-------|-------|-------|-------|---------|
> | Sentences Number Satisfaction Rate | 74.50 | 64.67 | 58.67 | 56.5  | 56.33   |
> | Instruction Satisfaction Rate| 42.83 | 40.50 | 32.67 | 31.33 | 10.17   |

---

> ### Author Response · Authors · 2024-11-20
>
> 3. Word counting: This example is obtained through GPT4-o with zero-shot. This shows that we can easily obtain reliable tools. The details are as follows:
>
> ```python
> def feedback(response, max_words=None, min_words=None):
>     # Count the number of words in the response
>     word_count = len(response.split())
>
>     # Check for maximum word constraint
>     if max_words is not None and word_count > max_words:
>         return f"Response failed because it has {word_count} words, exceeding the maximum allowed limit of {max_words} words."
>
>     # Check for minimum word constraint
>     if min_words is not None and word_count < min_words:
>         return f"Response failed because it has only {word_count} words, fewer than the minimum required {min_words} words."
>
>     # If all constraints are satisfied
>     return True
> ```
>
> ### Verification for New Constraints
>
> We appreciate the reviewer’s concern regarding the lack of existing tools for addressing new constraints.
> 1. ***Verification for New Constraints:*** For new constraints like generating text in a "Gothic" style, verification can be achieved if humans can evaluate the response. (a) Human judgments can be used to create training data, enabling the development of a “verifier” model that critiques responses. (b) Or we can train a small text-style classifier as a lightweight tool. To satisfy a completely new constraint, we need extra information about the constraint (through external tools or data).
> 2. ***Task Simplification:*** The DVR framework transforms the challenge of ensuring generation adherence (difficult task) to constraints into a verification task (simple task), simplifying the problem. If LLMs struggle to meet the new constraint and no verification tools or even data are available, we doubt if satisfying the constraint is a solvable problem.
> 3. Compared with regular fine-tuning methods, DVR provides an alternative solution (tool feedback) when there is no data for new constraints.

---

> ### Author Response · Authors · 2024-11-20
>
> ###  Evaluation on Each Module
>
> DVR consists of three components: **D**ecomposition & Tool Selection, **V**erification, and **R**efinement.
>
> 1. **Decomposition &Tool selection:** The instruction is first decomposed into single constraints, and the LLM selects the corresponding tool for each constraint. Tool selection accuracy is reported in **Table 5** and discussed in **Section 4.6 TOOL SELECTION ACCURACY** in the paper. These results already consider potential errors in the decomposition.  While there are many ways to decompose an instruction—making direct evaluation challenging—we evaluate whether the correct tools are selected, which is the goal of decomposition. Results show that LLMs can effectively select the correct tools.
>
> 2. **Verification**: Tools are carefully crafted and reliable. They give feedback based on the response. The tool example is shown above. **Refinement:** As shown in **Table 1, Table 2, and Figure 3** in the paper, the effectiveness of the refinement process can be observed by comparing the performance of vanilla outputs to those from DVR.
>
> 3. The experimental setting is fair for all baselines. Details are explained in paper 4.1 EXPERIMENTAL SETUP.
>
> 4. We would appreciate clarification on any specific details of the experiments that need further explanation.

---

> ### Author Response · Authors · 2024-11-23
>
> Thank you again for your valuable reviews and comments!
> We have updated our paper based on your suggestions. Specifically:
>
> 1. **Tool Correctness:**  The experiment with tool errors is added at **A.9 ROBUSTNESS OF DVR**. The discussion of the tool shortcoming is elaborated at **A.11 LIMITATIONS AND FUTURE WORKS**.
>
> 2. **Tool Examples:** Tool examples are added at **A.15 TOOL EXAMPLES**.
>
> The updated parts are all marked as blue so you can easily find them.
> If you have any further questions, please feel free to ask—we are happy to respond!

---

> ### Author Response · Authors · 2024-11-26
> **Reminder: Follow-Up on Rebuttal**
>
> Reviewer 6oxA:
>
> Regarding the weaknesses you pointed out, we have done our best to address them within the limited time available. As such, we would greatly appreciate it if you could review our responses and let us know if we have successfully addressed your concerns. If not, please feel free to share any additional feedback or remaining issues. Thanks.

---

> > ### Comment · Reviewer_6oxA · 2024-11-27
> > **Reply to authors**
> >
> > Thanks for your feedback, I decide to increase my score!

---

> > > ### Author Response · Authors · 2024-11-28
> > >
> > > Thank you for increasing the score. We appreciate your valuable feedback.
> > > Hope you have a great Thanksgiving!

---

### Official Review · Reviewer_UvT5 · 2024-10-31

**Soundness:** 3
**Presentation:** 2
**Contribution:** 2
**Rating:** 3
**Confidence:** 4

**Summary:**

In this paper, the authors propose *Divide-Verify-Refine (DVR)*, a framework designed to improve the adherence of large language models (LLMs) to complex, multi-constraint instructions, which divides instructions into single constraints, employs external tools for reliable verification and refines responses using a repository of successful adjustments. The authors conduct experiments on a new dataset to demonstrate the DVR's ability to enhance constraint adherence across various instruction types without requiring retraining.

**Strengths:**

The framework’s approach to decomposing complex instructions and using external tools to verify constraint adherence is novel in the context of LLM alignment without model retraining.

Evaluation results demonstrate DVR's effectiveness across varying constraint complexities.

The paper is clear, with well-defined modules for dividing, verifying, and refining stages, making the strategy understandable and reproducible.

**Weaknesses:**

The criteria for selecting external tools lack specificity, making it unclear how LLMs autonomously match tools to specific constraints.

The reliance on specific tools (e.g., Python-based scripts) could limit DVR’s generalizability across broader application domains or languages.

While ComplexInstruct is valuable, further validation on industry-standard benchmarks could better position DVR’s real-world applicability.

**Questions:**

Could the authors elaborate on how the DVR framework ensures that the refinement repository remains free from erroneous examples that could degrade LLM performance?

How might DVR handle instructions with unstructured or conflicting constraints, and are there plans to address such limitations in future work?

Could further details on how tools were evaluated for their accuracy and reliability in feedback generation be provided?

---

> ### Author Response · Authors · 2024-11-21
>
> We thank Reviewer UvT5 for their valuable comments. Below, we address each of your points individually.
>
> ## Tool Selection
>
> We appreciate the reviewer pointing out this problem. (a) We add the prompt that is used for LLM to select the tool based on the constraint. (b) Given the prompt, we check the response of LLMs to the list of tool names. If the response does not match any tool names, we will retry or drop this constraint after 5 trials.
>
> **Note**: The selection performance is reported in Table 5 and discussed in Section 4.6 in the paper.
>
> The selection prompt:
> ```
> You will be given a list of constraints. Each constraint belongs to a specific category. Your task is to recognize and categorize each constraint. Only output the category from the following options:
> postscript, placeholder, include keyword, exclude keyword, letter frequency, keyword frequency, sentence count constraint, word count constraint, *** separator, bullet points, fixed responses, highlighted, JSON format, title format, quoted response, end phrase, no commas, all capital letters, all lowercase, capital word frequency, language restriction
> Please ensure to categorize each constraint accurately according to its description. Here are examples for each type of constraint:
> Prompt: End it with a post script starting with P.S.
> Category: postscript
> Prompt: Limit the number of words you use to fewer than 65 words.
> Category: word count constraint
> Prompt: Make sure to include the word 'mutations'.
> Category: include keyword
> (more examples).....
> Prompt: {Current constraint}
> Category: {LLM Generation}
> ```
>
> Toolset= [ "postscript", “word count constraint”, "include keyword".....]
>
> We use difflib library's *get_close_matches* function, which identifies the closest match based on a similarity metric (Levenshtein Distance). Code:
>
> ```python
> import difflib
> def find_best_match(LLM_output,Toolset):
>     closest_match = difflib.get_close_matches(LLM_output, Toolset, n=1)
>     if closest_match:
>         return closest_match[0]
>     else:
>         return None
> ```
>
>
> ## Generalizability of DVR
>
> We appreciate the reviewer's concern regarding the reliance on specific tools.
>
> 1. **Tools are General:** The tools in our setting are not limited to codes. They can be open-source models. They can also be APIs that widely exist in the real world [1].
>
> 2. **Tool Making:** (a) New tools (python scripts) can be generated by advanced models like GPT-4. Tool generation is a one-time cost. These tools can be saved and used by cheap open-source models. Tool making is also a topic studied in ICLR24 [2]. (b) If we meet new constraints without existing tools and the constraint can not be verified by codes, verification can be achieved if humans can evaluate the response. Human judgments can be used to create training data, enabling the development of a “verifier” model that critiques responses.
>
> 4. **Domain Specialized Tools** In other specialized domains, we may apply or train specialized tools in that domain. For example, code generation tasks would have constraints (syntax or efficiency requirements), as mentioned by **Reviewer rh4y**. Tools may be interpreters or code analysis tools.
>
> **In conclusion, DVR actually offers greater generalizability compared to regular fine-tuning methods:** (a) Without domain data, DVR can use existing tools. (b) With domain data, we can try to train a critique tool or "domain expert" to provide feedback.
>
> [1] Qu, Changle, et al. "Tool Learning with Large Language Models: A Survey." 2024.
> [2] Cai, Tianle, et al. *"Large Language Models as Tool Makers."* ICLR 2024.
>
>
> ## Industry-standard Benchmarks
>
> 1. We did experiments on CoDI [1] an open-source benchmark. Results are shown in Table 3 in the paper.
> 2. We add experiments on IFEval [2] which is an instruction-following benchmark widely used for industry. The IFEval dataset evaluates the instruction-following ability and is one of the core benchmarks used in the ***Open LLM Leaderboard (Hugging Face)***. We conducted experiments on Mistral-7B-v0.3 and the results are shown below:
>
>
> Table 1. ISR (Instruction Satisfaction Ratio) for IFEval Dataset
> | Vanilla | Reflexion | BSM   | U-SC  | Rejection Sample | ReAct  | CRITIC | DVR   |
> |---------|-----------|-------|-------|--------------------|--------|--------|-------|
> | 47.32   | 47.13     | 47.87 | 46.95 | 53.23             | 53.97  | 55.53  | 60.44 |
>
> DVR still maintains the best performance compared with other baselines.
>
> [1] Chen, Yihan, et al. "Benchmarking large language models on controllable generation under diversified instructions." AAAI, 2024.
> [2] Zhou, Jeffrey, et al. "Instruction-following evaluation for large language models." arXiv,2023.

---

> ### Author Response · Authors · 2024-11-21
>
> ## Free from Erroneous Examples
>
> Each refinement process is saved only if the original response violates the constraint and the refined one satisfies the constraint. The tools essentially serve as gatekeepers in this process.
>
>
> ## Conflicting Constraints
>
> 1. This is a very good question. Currently, DVR does not consider conflicts between constraints since each constraint is managed independently. ***DVR iteratively selects the response that satisfies the highest number of constraints.***
> 2. ***However, managing conflicting constraints is an important and promising direction.*** Especially, if we have constraints in the system prompt. For instance, if a user's constraints conflict with those specified in the system prompt, we want the LLM to prioritize and adhere to the system prompt constraints. Prioritizing system constraints can improve the security and reliability of LLM systems, making this a valuable direction for future research.
>
> ## Evaluation on Tools
>
> 1. Tools in our experiments are checked by humans.
> 2. **New tools:** The quality of new tools can be checked through several approaches. There are techniques such as developing test cases that verify a tool’s outputs against known values [1,2] or generating multiple versions of a tool and selecting the most consistent output. For open-source models, quality can be checked through documentation and reported accuracy of models.
>
> [1] Cai, Tianle, et al. "Large Language Models as Tool Makers." ICLR 2024.
> [2] Huang, Dong, et al. "Agentcoder: Multi-agent-based code generation with iterative testing and optimisation." arXiv:2312.13010 (2023).
>
>
> 3. **Influence of Potential Errors:** We understand reviewer has concern on tool reliability. Although improving the quality of tools is out of the scope of our paper, we also conduct experiments to assess DVR’s performance in the presence of tool errors. Two types of errors are introduced: random noise and systematic bias. Specifically, we evaluate the framework on instructions with length constraints, using 600 samples (from ComplexInstruct) for word count control and another 600 for sentence count control. A constraint example: “The response needs to be less than (or at least) x number of words/sentences.” where x ranges from 10 to 100 for words and 3 to 5 for sentences.  We add two types of noises to tools: **Noise**: Gaussian noise with a mean of 0 is added to the counted number of words (or sentences) to simulate random errors. The DVR’s performance is then measured across different deviation levels. **Bias Errors**: A fixed bias is added to the counted values of words (or sentences) to introduce systematic errors. The tables below demonstrate DVR’s performance under different bias values.
>
> **Observations:** We have several observations in Table 1 and Table 2. (1) The performance will decrease as the noise levels (deviation, bias values) increase. (2) As the errors become large, the performance degradation will saturate. (3) Overall, DVR will not perform much worse than vanilla even if the bias and errors are large (20 for word count and 4 for sentence count). (4) The impact of noise on the overall instruction satisfaction rate is less severe compared to its influence on specific constraints.
>
> **Table 1. Satisfaction Rate (%) for word number constraints**
>
> | Deviation | 0    | 5    | 10    | 20    | Vanilla |
> |-----------|------|------|-------|-------|---------|
> | Words Number Satisfaction Rate | 88.00 | 87.17 | 82.83 | 81.50 | 68.16   |
> | Instruction Satisfaction Rate  | 48.17 | 45.00 | 43.67 | 43.67 | 10.17   |
>
> | Bias      | 0    | 5    | 10    | 20    | Vanilla |
> |-----------|------|------|-------|-------|---------|
> | Words Number Satisfaction Rate | 88.00 | 87.17 | 84.33 | 82.67 | 68.16   |
> | Instruction Satisfaction Rate| 48.17 | 47.83 | 47.00 | 45.67 | 10.17   |
>
> ---
>
> **Table 2. Satisfaction Rate for Sentence Number Constraints**
>
>
> | Deviation                          | 0     | 1     | 2     | 4     | Vanilla |
> |------------------------------------|-------|-------|-------|-------|---------|
> | Sentences Number Satisfaction Rate | 74.50 | 68.17 | 62.50 | 60.50 | 56.33   |
> | Instruction Satisfaction Rate | 42.83 | 38.17 | 35.33 | 34.33 | 10.17   |
>
> | Bias                               | 0     | 1     | 2     | 4     | Vanilla |
> |------------------------------------|-------|-------|-------|-------|---------|
> | Sentences Number Satisfaction Rate | 74.50 | 64.67 | 58.67 | 56.5  | 56.33   |
> | Instruction Satisfaction Rate| 42.83 | 40.50 | 32.67 | 31.33 | 10.17   |

---

> ### Author Response · Authors · 2024-11-21
>
> ### Example of external tools
>
> 1. Word counting: This example is obtained through GPT4-o with zero-shot. This shows that we can easily obtain reliable tools. The details are as follows:
>
> ```python
> def feedback(response, max_words=None, min_words=None):
>     # Count the number of words in the response
>     word_count = len(response.split())
>
>     # Check for maximum word constraint
>     if max_words is not None and word_count > max_words:
>         return f"Response failed because it has {word_count} words, exceeding the maximum allowed limit of {max_words} words."
>
>     # Check for minimum word constraint
>     if min_words is not None and word_count < min_words:
>         return f"Response failed because it has only {word_count} words, fewer than the minimum required {min_words} words."
>
>     # If all constraints are satisfied
>     return True
> ```
> 2.  Bullet points counting
>
> ```python
>
> class BulletList:
>     def __init__(self,num_bullets):
>         self._num_bullets = num_bullets
>     def feed_back(self, value):
>         r"""Check if the number of bullet lists meets the requirement.
>         Args:
>           value: A string representing the response. The response is expected to
>             contain some bullet lists that start with `\*`.
>
>         Returns:
>           True if the actual number of bullet lists in the response meets the
>           requirement.
>         """
>         bullet_lists = re.findall(r"^\s*\*[^\*].*$", value, flags=re.MULTILINE)
>         bullet_lists_2 = re.findall(r"^\s*-.*$", value, flags=re.MULTILINE)
>
>
>         num_bullet_lists = len(bullet_lists) + len(bullet_lists_2)
>
>         if num_bullet_lists == self._num_bullets:
>             return True
>         else:
>             return self.error_message(num_bullet_lists,bullet_lists,bullet_lists_2)
>
>     def error_message(self,num_bullet_lists,bullet_lists,bullet_lists_2):
>         # Combine both types of bullet points into a single list
>         all_bullet_lists = bullet_lists + bullet_lists_2
>         truncated_bullets = []
>         if len(all_bullet_lists) !=0:
>             for bullet in all_bullet_lists:
>                 bullet = bullet.strip()
>                 if len(bullet) > 30:
>                     truncated_bullets.append(bullet[:40] + "...")
>                 else:
>                     truncated_bullets.append(bullet)
>
>             # Create a summary of the bullets
>             bullet_summary = "\n".join(truncated_bullets)
>             # Construct the final error message
>             if len(all_bullet_lists) > self._num_bullets:
>                 add_num =len(all_bullet_lists)-self._num_bullets
>                 error_message = (
>                     f"In the response, there are {num_bullet_lists} bullet points.\n"
>                     f"Here are the bullet points detected:\n{bullet_summary}\n"
>                     f"Please remove exactly {add_num} bullet points to meet the requirement of {self._num_bullets}."
>                 )
>             else:
>                 shortage = self._num_bullets - num_bullet_lists
>                 error_message = (
>                     f"In the response, there are {num_bullet_lists} bullet points.\n"
>                     f"Here are the bullet points detected:\n{bullet_summary}\n"
>                     f"This is {shortage} fewer than needed. Please add exactly {shortage} more bullet points to meet the requirement of {self._num_bullets}."
>                 )
>         else:
>             error_message = (
>                 f"In the response, there is no bullet points.\n"
>                 f"We need exactly {self._num_bullets} number of bullet points."
>             )
>         return error_message
> ```

---

> ### Author Response · Authors · 2024-11-23
>
> Thank you again for your valuable reviews and comments!
> We have updated our paper based on your suggestions. Specifically:
>
> 1. **Tool Selection**: Selection prompt already exists at **A.13 PROMPTS**. The selection performance is reported in **Table 5** and discussed in **Section 4.6** in the paper.
>
> 2. **Industry-standard Benchmarks:** The experiment is added at **A.7 EXPERIMENTS ON IFEVAL**.
>
> 3. **Tool Examples:** Tool examples are added at **A.15 TOOL EXAMPLES**.
>
> 4. **Tool Correctness:**  The experiment with tool errors is added at **A.9 ROBUSTNESS OF DVR**. The discussion of the tool shortcoming is elaborated at **A.11 LIMITATIONS AND FUTURE WORKS**.
>
> The updated parts are all marked as blue so you can easily find them.
> If you have any further questions, please feel free to ask—we are happy to respond!

---

> ### Author Response · Authors · 2024-11-26
> **Reminder: Follow-Up on Rebuttal**
>
> Reviewer UvT5:
>
> Regarding the weaknesses you pointed out, we have done our best to address them within the limited time available. As such, we would greatly appreciate it if you could review our responses and let us know if we have successfully addressed your concerns. If not, please feel free to share any additional feedback or remaining issues. Thanks.

---

> ### Author Response · Authors · 2024-11-30
> **Reminder: Rebuttal Period Closing Soon**
>
> Reviewer UvT5:
>
> **As the rebuttal period will end soon**, we would greatly appreciate it if you could review our responses before the period closes and let us know if we have successfully addressed your concerns. If not, please feel free to share any additional feedback or remaining issues.
>
> Thanks.

---

> ### Author Response · Authors · 2024-12-03
> **Reminder: Rebuttal Period Closing Today**
>
> Reviewer UvT5:
>
> We have sent several reminders. **As the rebuttal period will end soon**, we would greatly appreciate it if you could review our responses before the period closes and let us know if we have successfully addressed your concerns.
>
> Thanks.

---

### Official Review · Reviewer_HS2T · 2024-11-02

**Soundness:** 3
**Presentation:** 2
**Contribution:** 2
**Rating:** 6
**Confidence:** 4

**Summary:**

Paper introduces Divide-Verify-Refine framework to overcome the struggele of LLMs for following instruction. In this framework first break down complex instructions into individual constraints and prepare suitable tools (Divide). Then it uses these tools to check responses and ensure quality feedback (Verify) and then it create a refinement repository to gather successful processes, using them as examples for future cases to improve the model (Refine). Also, authors develop new dataset of instructions, each containing 1-6
constraint.

**Strengths:**

The DVR framework introduces a structured methodology to process instructions, decomposing, verifying, and refining which can lead to better results.
The method is on the fly and not need for fine-tunning
DVR improves the performance across multiple datasets

**Weaknesses:**

The effectiveness of the system is on the quality and accuracy of the external tools used for verification and feedback. Poorly performing tools can degrade the overall performance of the LLM.

It would be beneficial to compare DVR method with other methods, such as ReAct, in other tasks such as  reasoning tasks.

**Questions:**

Take a look at the weakness section

---

> ### Author Response · Authors · 2024-11-21
>
> We appreciate reviewer's comments and suggestions. We will address your concerns in detail below.
>
> ## Tool Quality
>
> 1. We thank the reviewer for highlighting the concern regarding the reliability of newly generated tools. (a) However, it is essential to clarify that ensuring the correctness of these tools falls outside the scope of our study, as it relates more closely to the code-generation field, which have been extensively studied such as [1,2,3,4]. The tool making and verification process are also studied in ICLR24 [1]. There are techniques such as developing test cases that verify a tool’s outputs against known values [1,2] or generating multiple versions of a tool and selecting the most consistent output. (b) The “tool” in our setting is not limited to codes. They can also be APIs that widely exist in the real world [5]. If the tool we need does not exist, we can even train a tool if we can obtain data. (c) The tools used to verify responses are overall simple and easy to develop (e.g., counting words code). Down below, we also show an example of a tool obtained from GPT-4o with zero-shot. These tools are easy to develop, but LLM struggles to do the same job by itself (e.g., counting words). (d) In many real-world scenarios, constraints are predefined, allowing high-quality tools to be pre-prepared. For instance, in legal or technical writing, tools for format validation or terminology checks can be readily established.
>
> [1] Cai, Tianle, et al. *"Large Language Models as Tool Makers."* ICLR 2024.
> [2] Huang, Dong, et al. *"Agentcoder: Multi-agent-based code generation with iterative testing and optimisation."* arXiv:2312.13010 (2023).
> [3] Zhang, Shun, et al. "Planning with Large Language Models for Code Generation." ICLR 2023.
> [4] Jiang, Xue, et al. "Self-planning code generation with large language models." ACM Transactions on Software Engineering and Methodology.
> [5] Qu, Changle, et al. *"Tool Learning with Large Language Models: A Survey."* arXiv:2405.17935 (2024).
>
> 2. **Influence of Potential Errors** Although improving the quality of tools is out of the scope of our paper, we also conduct experiments to assess DVR’s performance in the presence of tool errors. Two types of errors are introduced: random noise and systematic bias. Specifically, we evaluate the framework on instructions with length constraints, using 600 samples (from ComplexInstruct) for word count control and another 600 for sentence count control. A constraint example: “The response needs to be less than (or at least) x number of words/sentences.” where x ranges from 10 to 100 for words and 3 to 5 for sentences.  We add two types of noises to tools: **Noise**: Gaussian noise with a mean of 0 is added to the counted number of words (or sentences) to simulate random errors. The DVR’s performance is then measured across different deviation levels. **Bias Errors**: A fixed bias is added to the counted values of words (or sentences) to introduce systematic errors. The tables below demonstrate DVR’s performance under different bias values.
>
> **Observations:** We have several observations in Table 1 and Table 2. (1) The performance will decrease as the noise levels (deviation, bias values) increase. (2) As the errors become large, the performance degradation will saturate. (3) Overall, DVR will not perform much worse than vanilla even if the bias and errors are large (20 for word count and 4 for sentence count). (4) The impact of noise on the overall instruction satisfaction rate is less severe compared to its influence on specific constraints.
>
> **Table 1. Satisfaction Rate (%) for word number constraints**
>
> | Deviation | 0    | 5    | 10    | 20    | Vanilla |
> |-----------|------|------|-------|-------|---------|
> | Words Number Satisfaction Rate | 88.00 | 87.17 | 82.83 | 81.50 | 68.16   |
> | Instruction Satisfaction Rate  | 48.17 | 45.00 | 43.67 | 43.67 | 10.17   |
>
> | Bias      | 0    | 5    | 10    | 20    | Vanilla |
> |-----------|------|------|-------|-------|---------|
> | Words Number Satisfaction Rate | 88.00 | 87.17 | 84.33 | 82.67 | 68.16   |
> | Instruction Satisfaction Rate| 48.17 | 47.83 | 47.00 | 45.67 | 10.17   |
>
> ---
>
> **Table 2. Satisfaction Rate for Sentence Number Constraints**
>
>
> | Deviation                          | 0     | 1     | 2     | 4     | Vanilla |
> |------------------------------------|-------|-------|-------|-------|---------|
> | Sentences Number Satisfaction Rate | 74.50 | 68.17 | 62.50 | 60.50 | 56.33   |
> | Instruction Satisfaction Rate | 42.83 | 38.17 | 35.33 | 34.33 | 10.17   |
>
> | Bias                               | 0     | 1     | 2     | 4     | Vanilla |
> |------------------------------------|-------|-------|-------|-------|---------|
> | Sentences Number Satisfaction Rate | 74.50 | 64.67 | 58.67 | 56.5  | 56.33   |
> | Instruction Satisfaction Rate| 42.83 | 40.50 | 32.67 | 31.33 | 10.17   |

---

> ### Author Response · Authors · 2024-11-21
>
> ### Example of external tools
>
> 1. Word counting: This example is obtained through GPT4-o with zero-shot. This shows that we can easily obtain reliable tools. The details are as follows:
>
> ```python
> def feedback(response, max_words=None, min_words=None):
>     # Count the number of words in the response
>     word_count = len(response.split())
>
>     # Check for maximum word constraint
>     if max_words is not None and word_count > max_words:
>         return f"Response failed because it has {word_count} words, exceeding the maximum allowed limit of {max_words} words."
>
>     # Check for minimum word constraint
>     if min_words is not None and word_count < min_words:
>         return f"Response failed because it has only {word_count} words, fewer than the minimum required {min_words} words."
>
>     # If all constraints are satisfied
>     return True
> ```
> 2.  Bullet points counting
>
> ```python
>
> class BulletList:
>     def __init__(self,num_bullets):
>         self._num_bullets = num_bullets
>     def feed_back(self, value):
>         r"""Check if the number of bullet lists meets the requirement.
>         Args:
>           value: A string representing the response. The response is expected to
>             contain some bullet lists that start with `\*`.
>
>         Returns:
>           True if the actual number of bullet lists in the response meets the
>           requirement.
>         """
>         bullet_lists = re.findall(r"^\s*\*[^\*].*$", value, flags=re.MULTILINE)
>         bullet_lists_2 = re.findall(r"^\s*-.*$", value, flags=re.MULTILINE)
>
>
>         num_bullet_lists = len(bullet_lists) + len(bullet_lists_2)
>
>         if num_bullet_lists == self._num_bullets:
>             return True
>         else:
>             return self.error_message(num_bullet_lists,bullet_lists,bullet_lists_2)
>
>     def error_message(self,num_bullet_lists,bullet_lists,bullet_lists_2):
>         # Combine both types of bullet points into a single list
>         all_bullet_lists = bullet_lists + bullet_lists_2
>         truncated_bullets = []
>         if len(all_bullet_lists) !=0:
>             for bullet in all_bullet_lists:
>                 bullet = bullet.strip()
>                 if len(bullet) > 30:
>                     truncated_bullets.append(bullet[:40] + "...")
>                 else:
>                     truncated_bullets.append(bullet)
>
>             # Create a summary of the bullets
>             bullet_summary = "\n".join(truncated_bullets)
>             # Construct the final error message
>             if len(all_bullet_lists) > self._num_bullets:
>                 add_num =len(all_bullet_lists)-self._num_bullets
>                 error_message = (
>                     f"In the response, there are {num_bullet_lists} bullet points.\n"
>                     f"Here are the bullet points detected:\n{bullet_summary}\n"
>                     f"Please remove exactly {add_num} bullet points to meet the requirement of {self._num_bullets}."
>                 )
>             else:
>                 shortage = self._num_bullets - num_bullet_lists
>                 error_message = (
>                     f"In the response, there are {num_bullet_lists} bullet points.\n"
>                     f"Here are the bullet points detected:\n{bullet_summary}\n"
>                     f"This is {shortage} fewer than needed. Please add exactly {shortage} more bullet points to meet the requirement of {self._num_bullets}."
>                 )
>         else:
>             error_message = (
>                 f"In the response, there is no bullet points.\n"
>                 f"We need exactly {self._num_bullets} number of bullet points."
>             )
>         return error_message
> ```

---

> ### Author Response · Authors · 2024-11-21
>
> ## Additional Experiments
>
> Thank you for your suggestions. (1) We believe following complex instructions is already a reasoning-intensive task. (2) We add experiments on IFEval [1] which is an instruction-following benchmark widely used for industry. The IFEval dataset evaluates the instruction-following ability and is one of the core benchmarks used in the ***Open LLM Leaderboard (Hugging Face)***. We conducted experiments on Mistral-7B-v0.3 and the results are shown below:
>
>
> Table 1. ISR (Instruction Satisfaction Ratio) for IFEval Dataset
> | Vanilla | Reflexion | BSM   | U-SC  | Rejection Sample | ReAct  | CRITIC | DVR   |
> |---------|-----------|-------|-------|--------------------|--------|--------|-------|
> | 47.32   | 47.13     | 47.87 | 46.95 | 53.23             | 53.97  | 55.53  | 60.44 |
>
> DVR still maintains the best performance compared with other baselines.
>
> [1] Zhou, Jeffrey, et al. "Instruction-following evaluation for large language models." arXiv,2023.
>
> (3) We will try to find more reasoning datasets and run experiments if time allows. We appreciate it if you could suggest specific datasets.

---

> ### Author Response · Authors · 2024-11-23
>
> Thank you again for your valuable reviews and comments!
> We have updated our paper based on your suggestions. Specifically:
>
>
> 1. **Tool Quality:**  The experiment with tool errors is added at **A.9 ROBUSTNESS OF DVR**. The discussion of the tool shortcoming is elaborated at **A.11 LIMITATIONS AND FUTURE WORKS**.
>
> 2. **Tool Examples:** Tool examples are added at **A.15 TOOL EXAMPLES**.
>
> 3. **Additional Experiment:** The experiment on IFeval is added at **A.7 EXPERIMENTS ON IFEVAL**.
>
> The updated parts are all marked as blue so you can easily find them.
> If you have any further questions, please feel free to ask—we are happy to respond!

---

### Official Review · Reviewer_rh4y · 2024-11-03

**Soundness:** 4
**Presentation:** 4
**Contribution:** 3
**Rating:** 8
**Confidence:** 4

**Summary:**

This paper proposes a technique called DVR to improve the alignment of LLM responses specifically with the constraints given within the responses. The technique is a three stage iterative process, where the first stage gathers the constraints, the second gets the response from the LLM, and the third iteratively refines the response based on the constraints satisfied. The technique improves the performance of major LLM models by a significant amount.

**Strengths:**

Firstly, I really liked how the paper was written. It was clear and easy to understand. The problem being addressed is a well documented issue of response alignment, and the technique seems to use a divide and conquer approach to solve it, which seems quite novel. The paper also presents a thorough evaluation of the technique over multiple models compared with several baselines.

**Weaknesses:**

I don't see too many weaknesses in the paper. One thing I would like to see addressed is a limitations or future work section to specify the shortcomings of the tool. It would also be nice to have a discussion on what it would take to expand the constraints from being purely conjunctive (which I think it is right now) to including disjunctions and negations at the very least. I also think it is important to better showcase that in each iteration of the feedback generation, an LLM is called to get the refined response. Right now when I look at the diagram, unless I am paying attention to the arrow shapes, I would think that the LLM is only called when getting the initial response.

**Questions:**

1. Is this system modular? I.e. what would it take for us to incorporate additional kinds of constraints, tools, etc.?
2. Are there any other datasets you can include? Maybe datasets like code generation datasets where constraints can be more implicit but can be programmed into the prompt from the evaluator's side (like syntax correctness, correct return types, etc.)? This is dependent on the amount of effort required to get this working with such datasets.
3. Is there a reason you don't evaluate GPT models? I understand that such errors are more prone to occur in open source models but it would be nice to see GPT-4 as a baseline

---

> ### Author Response · Authors · 2024-11-21
>
> We appreciate the reviewer’s recognition of the significance of our work. We will address reviewer's concern one by one.
>
> ## Shortcomings of the Tools
>
> We appreciate the reviewer can point out this problem. We will include and discuss this in our paper.
>
> **Missing Tools:** Currently, DVR can only satisfy constraints that have corresponding tools. When we meet new constraints, new tools are needed to verify that constraint. There are several situations: (a) New tools (python scripts) can be generated by advanced models like GPT-4. Tool generation is a one-time cost. These tools can be saved and used by cheap open-source models. Tool making is also a topic studied in ICLR24 [1]. (b) New tools can also be open-source models and APIs [3]. (c) If we meet new constraints without existing tools and the constraint can not be verified by codes, verification can be achieved if humans can evaluate the response. Human judgments can be used to create training data, enabling the development of a “verifier” model that critiques responses.
>
> ## More Complex Constraints Relationships
>
> 1. This is a very good point. Actually, it is a future direction we try to explore.
> We want to deal with more complex instructions where constraints not only have “and” relationships but also “or” relationships. Constraints may also have **dependency** which means satisfying one constraint may need to satisfy its pre-requisite first.
> 2. **Potential Direction:** These complex relationships might be modeled as the “graph” structure. Under such scenario, the “Decomposition” part in DVR needs certain reasoning ability of LLM to analyze the dependency and logical relationships. For example, LLM may need to judge which constraint might be easier to satisfy when it meets disjunctions.
>
> ## Better Illustration
>
> Thank you for pointing out this problem. We will fix the diagram to better illustrate the process.
>
> ## Additional Constraints and Tools
>
> Yes. DVR is modular.
>
> **Tool Quality:** Considering we use LLMs to generate tools, tool quality becomes an important part. There are techniques such as developing test cases that verify a tool’s outputs against known values [1,2] or generating multiple versions of a tool and selecting the most consistent output.
>
>
>
> ## Additional Experiments
>
> 1. Challenge: Thank you for pointing out this problem. We have not found any coding datasets that have such features. Actually, we are very interested in this topic and we are now doing some surveys on code generation. Currently, the constraint-following topic is a new problem problem. This means there are only several works that make benchmarks for regular constraints (most of them are works in 2024).
>
> 2. **IFeval [4]:** We add experiments on IFEval which is an instruction-following benchmark widely used for industry. The IFEval dataset evaluates the instruction-following ability and is one of the core benchmarks used in the **Open LLM Leaderboard (Hugging Face)**. We conducted experiments on Mistral-7B-v0.3 and the results are shown below:
>
>
> Table 1. ISR (Instruction Satisfaction Ratio) for IFEval Dataset
> | Vanilla | Reflexion | BSM   | U-SC  | Rejection Sample | ReAct  | CRITIC | DVR   |
> |---------|-----------|-------|-------|--------------------|--------|--------|-------|
> | 47.32   | 47.13     | 47.87 | 46.95 | 53.23             | 53.97  | 55.53  | 60.44 |
>
> DVR still maintains the best performance compared with other baselines.
>
> [1] Cai, Tianle, et al. "Large Language Models as Tool Makers." ICLR 2024.
> [2] Huang, Dong, et al. "Agentcoder: Multi-agent-based code generation with iterative testing and optimisation." arXiv,2023.
> [3] Qu, Changle, et al. "Tool Learning with Large Language Models: A Survey." arXiv,2024.
> [4] Zhou, Jeffrey, et al. "Instruction-following evaluation for large language models." arXiv,2023.
>
>
> ## Evaluation on GPT-4-turbo
>
> Thank you for pointing out this problem, we add experiments on GPT-4-turbo. Shown in Table 1, we can observe that GPT-4-turbo performs better than open-source models (Mistral and Llama). Surprisingly, applied on Llama3.1-8B, DVR can still outperform GPT-4-turbo, indicating that DVR exploits the potential of the open-source model.
>
> ### Table 1. Instruction Satisfaction Ratio (%)
>
> | Model               | Level 1 | Level 2 | Level 3 | Level 4 | Level 5 | Level 6 |
> |---------------------|---------|---------|---------|---------|---------|---------|
> | Mistral-7B          | 77.0    | 55.3    | 34.1    | 19.9    | 12.4    | 6.3     |
> | DVR (Mistral-7B)    | 95.0    | 81.3    | 66.6    | 51.4    | 36.4    | 23.4    |
> | Llama3.1-8B         | 90.5    | 76.6    | 62.5    | 50.1    | 35.6    | 25.3    |
> | DVR (Llama3.1-8B)   | 95.2    | 88.7    | 79.2    | 69.7    | 60.5    | 49.6    |
> | GPT-4-turbo         | 95.3    | 88.4    | 78.8    | 65.2    | 53.7    | 42.6    |

---

> ### Author Response · Authors · 2024-11-23
>
> Thank you again for your valuable reviews and comments!
> We have updated our paper based on your suggestions. Specifically:
>
>
> 1. **Shortcomings of the Tools:**  The experiment with tool errors is added at **A.9 ROBUSTNESS OF DVR**. The discussion of the tool shortcoming is elaborated at **A.11 LIMITATIONS AND FUTURE WORKS**.
>
> 2. **Additional Experiment:** The experiment on IFeval is added at **A.7 EXPERIMENTS ON IFEVAL**.
>
> 3. **Tool Examples:** Tool examples are added at **A.15 TOOL EXAMPLES**.
>
> 4. **GPT-4-turbo:**  The experiment on GPT-4-turbo is at **A.8 EXPERIMENTS ON GPT4-TURBO**.
>
> The updated parts are all marked as blue so you can easily find them.
> If you have any further questions, please feel free to ask—we are happy to respond!

---

### Official Review · Reviewer_xMKj · 2024-11-04

**Soundness:** 3
**Presentation:** 3
**Contribution:** 1
**Rating:** 5
**Confidence:** 3

**Summary:**

This paper proposes Divide-Verify-Refine (DVR), which is a framework that helps LLM generate responses that can meet complex instructions. Specifically, DVR is divided into three steps: (1) Divide, where LLMs is prompted to divide instructions into multiple constraints and to prepare approximate tools for each constraint; (2) Verify, where tools are used to check whether the response meets corresponding constraints and, if not, provide detailed feedback; (3) Refine, where a refinement repository is proposed to collect successful refinement processes which will be used as few-shot examples when LLMs use the detailed tool feedbacks to refine their previous responses. Evaluation on CoDI and ComplexInstruct (a newly proposed benchmark) across different LLMs demonstrates the effectiveness of DVR.

**Strengths:**

- The paper proposes a new benchmark to better evaluate complex instruction-following capabilities of LLMs.
- The paper includes a comprehensive evaluation with a detailed analysis on the effectiveness of DVR.

**Weaknesses:**

- The primary concern with this work is its novelty. While authors claim that DVR differs from CRITIC by (1) incorporating a refinement repository module to provide few-shot examples and (2) using multiple tools rather than a single tool to provide a more detailed feedback, such differences are quite minimal and don't seem to be pass the bar for ICLR.
- The assumption of external tools being available for all existing constraints is not realistic. While authors claim that code generation models can be used to generate Python scripts (i.e., new tools) to check new constraints when there are no existing tools, the correctness of the newly generated tools cannot be guaranteed, and thus their detailed feedback will not be reliable.

**Questions:**

- Are there any examples of external tools (e.g., Python code) being used in the experiment?
- What is the overhead of DVR compared with all the baselines?

---

> ### Author Response · Authors · 2024-11-20
>
> We appreciate the reviewer xMKj comments. Below, we address your concerns one by one:
>
> ### Distinction from CRITIC
> DVR uses multiple specialized tools and has a different objective than CRITIC, focusing on handling complex instructions with multiple constraints. Its refinement repository allows LLMs to learn from past refinements, enhancing the refinement effectiveness. Additionally, DVR features a distinct pipeline and delivers better performance. The detailed distinctions are as follows:
>
> 1. **Distinct Objective and Specialized Tool Use:** Tools are used differently in our framework compared to CRITIC. For example, CRITIC utilizes ‘Wiki’ as a tool for Question Answering tasks to retrieve extra information, functioning more like Retrieval-Augmented Generation. In contrast, our work focuses on complex instructions that pose a unique challenge: it is difficult for LLMs to generate a single response satisfying multiple constraints, but these constraints can be easily verified individually using specialized tools (e.g., Python code). In this context, tools serve as a complement to the weaknesses of LLMs. DVR is specifically designed to address the complex instruction-following problem, which is distinct from the task in CRITIC.
> 2. **Maximizing Tool Effectiveness in Refinement:** Another key contribution of our work is to maximize the benefit of past refinement processes for future refinements. Instead of using fixed few-shot demonstrations, our framework innovatively designed a refinement repository module, which selectively provides examples that: (a) have been verified by tools to ensure the initial response fails the constraint, while the refined response satisfies it. (b) match the specific constraint type needing refinement in the current response. This constraint-type-based selection approach offers a **distinct view compared to traditional retrieval-augmented generation** by retrieving examples specifically aligned with the constraint type rather than relying solely on semantic similarity. As a result, the refinement repository enables DVR to be distinct from previous works.
> 3. **Distinct Pipeline and Significant Performance Gain:**
> Unlike CRITIC, which relies on a single, predefined tool for a task, DVR tackles instructions with multiple constraints. (a) DVR allows the LLM to decompose complex instructions into individual constraints and then select appropriate tools from a pool, rather than depending on a single fixed, predefined tool. (b) DVR provides detailed feedback that locates the error (e.g., There are 4 bullet points in the response. Here are the bullet points detected:.....) and provides directional information (e.g. Please remove exactly 1 bullet point to meet the requirement of 3) to guide LLM. From the experiment results, we can also observe that learning from past experience can improve performance. As shown by Table 1 and Table 2 in paper, DVR consistently outperforms CRITIC, especially on tasks with higher constraint counts (5-6 constraints).
>
> For these reasons, we believe DVR offers a novel solution for complex instruction-following problems and significantly enhances LLM performance.

---

> ### Author Response · Authors · 2024-11-20
>
> ### Tool Correctness
>
> 1. We thank the reviewer for highlighting the concern regarding the reliability of newly generated tools. (a) However, it is essential to clarify that ensuring the correctness of these tools falls outside the scope of our study, as it relates more closely to the code-generation field, which have been extensively studied such as [1,2,3,4]. The tool making and verification process are also studied in ICLR24 [1]. There are techniques such as developing test cases that verify a tool’s outputs against known values [1,2] or generating multiple versions of a tool and selecting the most consistent output. (b) The “tool” in our setting is not limited to codes. They can also be APIs that widely exist in the real world [5]. If the tool we need does not exist, we can even train a tool if we can obtain data. (c) The tools used to verify responses are overall simple and easy to develop (e.g., counting words code). Down below, we also show an example of a tool obtained from GPT-4o with zero-shot. These tools are easy to develop, but LLM struggles to do the same job by itself (e.g., counting words). (d) In many real-world scenarios, constraints are predefined, allowing high-quality tools to be pre-prepared. For instance, in legal or technical writing, tools for format validation or terminology checks can be readily established.
>
> [1] Cai, Tianle, et al. *"Large Language Models as Tool Makers."* ICLR 2024.
> [2] Huang, Dong, et al. *"Agentcoder: Multi-agent-based code generation with iterative testing and optimisation."* arXiv:2312.13010 (2023).
> [3] Zhang, Shun, et al. "Planning with Large Language Models for Code Generation." ICLR 2023.
> [4] Jiang, Xue, et al. "Self-planning code generation with large language models." ACM Transactions on Software Engineering and Methodology.
> [5] Qu, Changle, et al. *"Tool Learning with Large Language Models: A Survey."* arXiv:2405.17935 (2024).
>
> 2. **Influence of Potential Errors** Although improving the quality of tools is outthe scope of our paper, we also conduct experiments to assess DVR’s performance in the presence of tool errors. Two types of errors are introduced: random noise and systematic bias. Specifically, we evaluate the framework on instructions with length constraints, using 600 samples (from ComplexInstruct) for word count control and another 600 for sentence count control. A constraint example: “The response needs to be less than (or at least) x number of words/sentences.” where x ranges from 10 to 100 for words and 3 to 5 for sentences.  We add two types of noises to tools: **Noise**: Gaussian noise with a mean of 0 is added to the counted number of words (or sentences) to simulate random errors. The DVR’s performance is then measured across different deviation levels. **Bias Errors**: A fixed bias is added to the counted values of words (or sentences) to introduce systematic errors. The tables below demonstrate DVR’s performance under different bias values.
>
> **Observations:** We have several observations in Table 1 and Table 2. (1) The performance will decrease as the noise levels (deviation, bias values) increase. (2) As the errors become large, the performance degradation will saturate. (3) Overall, DVR will not perform much worse than vanilla even if the bias and errors are large (20 for word count and 4 for sentence count). (4) The impact of noise on the overall instruction satisfaction rate is less severe compared to its influence on specific constraints.
>
> **Table 1. Satisfaction Rate (%) for word number constraints**
>
> | Deviation | 0    | 5    | 10    | 20    | Vanilla |
> |-----------|------|------|-------|-------|---------|
> | Words Number Satisfaction Rate | 88.00 | 87.17 | 82.83 | 81.50 | 68.16   |
> | Instruction Satisfaction Rate  | 48.17 | 45.00 | 43.67 | 43.67 | 10.17   |
>
> | Bias      | 0    | 5    | 10    | 20    | Vanilla |
> |-----------|------|------|-------|-------|---------|
> | Words Number Satisfaction Rate | 88.00 | 87.17 | 84.33 | 82.67 | 68.16   |
> | Instruction Satisfaction Rate| 48.17 | 47.83 | 47.00 | 45.67 | 10.17   |
>
> ---
>
> **Table 2. Satisfaction Rate for Sentence Number Constraints**
>
> Table 2. Satisfaction Rate for sentence number constraints
>
> | Deviation                          | 0     | 1     | 2     | 4     | Vanilla |
> |------------------------------------|-------|-------|-------|-------|---------|
> | Sentences Number Satisfaction Rate | 74.50 | 68.17 | 62.50 | 60.50 | 56.33   |
> | Instruction Satisfaction Rate | 42.83 | 38.17 | 35.33 | 34.33 | 10.17   |
>
> | Bias                               | 0     | 1     | 2     | 4     | Vanilla |
> |-----------------------------------|-------|-------|-------|-------|---------|
> | Sentences Number Satisfaction Rate | 74.50 | 64.67 | 58.67 | 56.5  | 56.33   |
> | Instruction Satisfaction Rate| 42.83 | 40.50 | 32.67 | 31.33 | 10.17   |

---

> ### Author Response · Authors · 2024-11-20
>
> ### Example of external tools
>
> 1. Word counting: This example is obtained through GPT4-o with zero-shot. This shows that we can easily obtain reliable tools. The details are as follows:
>
> ```python
> def feedback(response, max_words=None, min_words=None):
>     # Count the number of words in the response
>     word_count = len(response.split())
>
>     # Check for maximum word constraint
>     if max_words is not None and word_count > max_words:
>         return f"Response failed because it has {word_count} words, exceeding the maximum allowed limit of {max_words} words."
>
>     # Check for minimum word constraint
>     if min_words is not None and word_count < min_words:
>         return f"Response failed because it has only {word_count} words, fewer than the minimum required {min_words} words."
>
>     # If all constraints are satisfied
>     return True
> ```
> 2.  Bullet points counting
>
> ```python
>
> class BulletList:
>     def __init__(self,num_bullets):
>         self._num_bullets = num_bullets
>     def feed_back(self, value):
>         r"""Check if the number of bullet lists meets the requirement.
>         Args:
>           value: A string representing the response. The response is expected to
>             contain some bullet lists that start with `\*`.
>
>         Returns:
>           True if the actual number of bullet lists in the response meets the
>           requirement.
>         """
>         bullet_lists = re.findall(r"^\s*\*[^\*].*$", value, flags=re.MULTILINE)
>         bullet_lists_2 = re.findall(r"^\s*-.*$", value, flags=re.MULTILINE)
>
>
>         num_bullet_lists = len(bullet_lists) + len(bullet_lists_2)
>
>         if num_bullet_lists == self._num_bullets:
>             return True
>         else:
>             return self.error_message(num_bullet_lists,bullet_lists,bullet_lists_2)
>
>     def error_message(self,num_bullet_lists,bullet_lists,bullet_lists_2):
>         # Combine both types of bullet points into a single list
>         all_bullet_lists = bullet_lists + bullet_lists_2
>         truncated_bullets = []
>         if len(all_bullet_lists) !=0:
>             for bullet in all_bullet_lists:
>                 bullet = bullet.strip()
>                 if len(bullet) > 30:
>                     truncated_bullets.append(bullet[:40] + "...")
>                 else:
>                     truncated_bullets.append(bullet)
>
>             # Create a summary of the bullets
>             bullet_summary = "\n".join(truncated_bullets)
>             # Construct the final error message
>             if len(all_bullet_lists) > self._num_bullets:
>                 add_num =len(all_bullet_lists)-self._num_bullets
>                 error_message = (
>                     f"In the response, there are {num_bullet_lists} bullet points.\n"
>                     f"Here are the bullet points detected:\n{bullet_summary}\n"
>                     f"Please remove exactly {add_num} bullet points to meet the requirement of {self._num_bullets}."
>                 )
>             else:
>                 shortage = self._num_bullets - num_bullet_lists
>                 error_message = (
>                     f"In the response, there are {num_bullet_lists} bullet points.\n"
>                     f"Here are the bullet points detected:\n{bullet_summary}\n"
>                     f"This is {shortage} fewer than needed. Please add exactly {shortage} more bullet points to meet the requirement of {self._num_bullets}."
>                 )
>         else:
>             error_message = (
>                 f"In the response, there is no bullet points.\n"
>                 f"We need exactly {self._num_bullets} number of bullet points."
>             )
>         return error_message
> ```
>
> ### Computation time
>
> We conducted experiments with 20 instructions, each containing 6 constraints, using Mistral-7B. The number of trials was set to 5, consistent with the paper's settings. The average running time is summarized below:
>
> Table 1. The average running time for one sample in seconds
>
> | Vanilla | Reflexion | U-SC  | BSM   | Rejection Sample | ReAct  | CRITIC | DVR   |
> |---------|-----------|-------|-------|--------------------|--------|--------|-------|
> | 5.91    | 20.53     | 41.34 | 46.21 | 32.32             | 36.48  | 37.98  | 33.91 |
>
> Table 2. The ISR of each method (copied from Table 2 in the paper)
>
> | Vanilla | Reflexion | U-SC  | BSM   | Rejection Sample | ReAct  | CRITIC | DVR   |
> |---------|-----------|-------|-------|--------------------|--------|--------|-------|
> | 6.3     | 5.8       | 5.2   | 5.8   | 6.8                | 10.7   | 18.1   | 23.6  |
>
>
> As shown in Table 1, our method does not exhibit significantly higher running time compared to other baselines. Considering the performance gains (Table 2), our method demonstrates a balance between efficiency and effectiveness.

---

> ### Author Response · Authors · 2024-11-23
>
> Thank you again for your valuable reviews and comments!
> We have updated our paper based on your suggestions. Specifically:
>
> 1. **Tool Correctness:**  The experiment with tool errors is added at **A.9 ROBUSTNESS OF DVR**. The discussion of the tool shortcoming is elaborated at **A.11 LIMITATIONS AND FUTURE WORKS**.
>
> 2. **Tool Examples:** Tool examples are added at **A.15 TOOL EXAMPLES**.
>
> 3. **Computation Time:** Experiment is added at **A.10 COMPUTATION TIME**
>
> The updated parts are all marked as blue so you can easily find them.
> If you have any further questions, please feel free to ask—we are happy to respond!

---

> ### Author Response · Authors · 2024-11-26
> **Reminder: Follow-Up on Rebuttal**
>
> Reviewer xMKj:
>
> Regarding the weaknesses you pointed out, we have done our best to address them within the limited time available. As such, we would greatly appreciate it if you could review our responses and let us know if we have successfully addressed your concerns. If not, please feel free to share any additional feedback or remaining issues. Thanks.

---

> ### Author Response · Authors · 2024-11-30
> **Reminder: Rebuttal Period Closing Soon**
>
> Reviewer xMKj:
>
> **As the rebuttal period will end soon**, we would greatly appreciate it if you could review our responses before the period closes and let us know if we have successfully addressed your concerns. If not, please feel free to share any additional feedback or remaining issues.
>
> Thanks.

---

> > ### Comment · Reviewer_xMKj · 2024-12-02
> >
> > Thanks for the additional results and discussion. I particularly appreciate your additional experiment on the influence of potential tool errors, so I decided to raise my score to 5.

---

### Meta-Review · Area_Chair_Cz2G · 2024-12-21

**Metareview:**

This paper proposes the Divide-Verify-Refine (DVR) framework for improving LLM's abilities to follow complex instructions. It first divides complex instructions into constraints which can be verified by existing tools, and uses a refinement repository (collecting successful updates) as few-shot examples for future queries. The authors perform experiments on CoDI and also build a new dataset (consisting of instructions with 1-6 constraints) called ComplexInstruct, using CoDI as seed instructions with added constraints. Results show that DVR is able to outperform baselines like ReAct and CRITIC on these datasets. Reviewers appreciated the problem of complex instruction following tackled by this paper as well as improving the LLM performance without re-training. However, concerns are raised on the contributions of this approach wrt CRITIC (ICLR 2024), tool correctness (e.g. what if tools make an error), as well as tool availability for potentially new tasks. This paper received mixed reviews after rebuttal - after taking a close look at the discussion and author responses, I think the biggest concern raised by almost all reviewers is tool availability and tool correctness. The authors argue that tool correctness is out of scope of the paper. However, concerns remain on tool availability (and the ability of LLMs to use these tools without retraining). This is because all the experiments focus on CoDI + an extension to CoDI introduced by the authors (focusing on constraints for text generation, like including keywords, length, or formatting) + IFEval during rebuttal (but seems to contain similar types of constraints). It is difficult to evaluate the generalizability of the results in this paper for other types of "complex instructions". For example, in CRITIC, tasks from multiple domains are considered (including QA, math, and toxicity). Therefore, experiments from additional domains would have helped demonstrate the significance of this method over CRITIC. Another experiment that would be helpful for strengthening the paper, would be a slightly improved CRITIC with divide - first use a LLM to divide the constraints, and call CRITIC on each constraint. How does this perform compared to DVR?

**Additional Comments On Reviewer Discussion:**

Reviewers initially raised concerns on novelty of the approach, tool correctness, tool availability, and more details on parts of the method (e.g. tool selection). Some concerns were addressed during rebuttal and the authors performed experiments on one additional dataset; a few reviewers raised their score (to 5). One of the main points to understand is whether this work is sufficiently significant for publication given existing works in this area - tool use for correcting LLM outputs have been studied by other baselines. While the authors do show better performance using DVR + the task is on more complex instructions, it seems to be a more limited domain compared to prior work, and it is unclear if DVR is generally stronger (for instructions on things besides keywords / formatting / word count, etc.). Also, adding experiments suggested above would be helpful to position this work compared to baselines like CRITIC. Another option is to re-frame this paper to focus on controllable text generation with this specific set of instruction list, since as is, I think the experiments are too narrow to back up the claims.

---

### Decision · Program_Chairs · 2025-01-22

Reject